# Net irrigation requirement under different climate scenarios using AquaCrop over Europe

Louise Busschaert[1], Shannon de Roos[1], Wim Thiery[2], Dirk Raes[1], Gabriëlle J. M. De Lannoy[1]

[1]Department of Earth and Environmental Sciences, KU Leuven, Heverlee, B-3001, Belgium
[2]Department of Hydrology and Hydraulic Engineering, Vrije Universiteit Brussel, Brussels, B-1050, Belgium

*Correspondence to*: Louise Busschaert (louise.busschaert@kuleuven.be)

**Abstract.** Global soil water availability is challenged by the effects of climate change and a growing population. On average 70% of freshwater extraction is attributed to agriculture, and the demand is increasing. In this study, the effects of climate change on the evolution of the irrigation water requirement to sustain current crop productivity are assessed by using the FAO

crop growth model AquaCrop version 6.1. The model is run at 0.5° lat x 0.5° lon resolution over the European mainland, assuming a general C3-type of crop, and forced by climate input data from the Inter-Sectoral Impact Model Intercomparison Project phase three (ISIMIP3).

First, the AquaCrop surface soil moisture (SSM) forced with two types of ISIMIP3 historical meteorological datasets is evaluated with satellite-based SSM estimates in two ways. When driven by ISIMIP3a reanalysis meteorology, daily simulated

SSM values have an unbiased root-mean-square difference of 0.08 and 0.06 $m^3m^{-3}$ with SSM retrievals from the Soil Moisture Ocean Salinity (SMOS) and Soil Moisture Active Passive (SMAP) missions, respectively, for the years 2015-2016 (2016 is the end year of the reanalysis data). When forced with ISIMIP3b meteorology from five Global Climate Models (GCM) for the years 2015-2020, the historical simulated SSM climatology closely agrees with the satellite-based SSM climatologies.

Second, the evaluated AquaCrop model is run to quantify the future irrigation requirement, for an ensemble of five GCMs and

three different emission scenarios. The simulated net irrigation requirement ($I_{net}$) of the three summer months for a near and far future climate period (2031-2060 and 2071-2100) is compared to the baseline period of 1985-2014, to assess changes in the mean and interannual variability of the irrigation demand. Averaged over the continent and the model ensemble, the far future $I_{net}$ is expected to increase by 22 mm month$^{-1}$ (+30%) under a high emission scenario Shared Socioeconomic Pathway (SSP) 3-7.0. Central and southern Europe are the most impacted with larger $I_{net}$ increases. The interannual variability of $I_{net}$ is

likely to increase in northern and central Europe, whereas the variability is expected to decrease in southern regions. Under a high mitigation scenario (SSP1-2.6), the increase in $I_{net}$ will stabilize around 13 mm month$^{-1}$ towards the end of the century and interannual variability will still increase but to a smaller extent. The results emphasize a large uncertainty in the $I_{net}$ projected by various GCMs.

**1 Introduction**

Global crop production has vastly increased over the past century, leading to the expansion of irrigated areas by almost sixfold, and more pressure on the irrigation water demand (Siebert et al., 2015). With changing climatic conditions and a growing population, future water availability is expected to further decline, raising demands for more efficient irrigation systems (Elliott et al., 2014; Taylor et al., 2013) and a higher crop water productivity (Brauman et al., 2021). In this context, a range of modelling studies have tried to assess future impacts on agricultural water demands and possible actions, but this remains a

difficult task due to high uncertainties in future climate and socioeconomic scenarios (Elliott et al., 2014; Haddeland et al., 2014; Wada et al., 2013).

Future meteorological variables are typically modelled by Global Climate Models (GCMs) for different scenarios, usually represented by the Representative Concentration Pathways (RCPs; van Vuuren et al., 2011). Some challenges associated with climate forcing data are the consistency and the representation of the uncertainty of the data. The Inter-Sectoral Impact Model

Intercomparison Project (ISIMIP) is an initiative to provide consistent bias-corrected climate datasets for impact modelling (Rosenzweig et al., 2017; Warszawski et al., 2014). The project is currently at its third simulation round (ISIMIP3) and provides reanalysis historical climate (ISIMIP3a) and GCM-driven historical and future climate (ISIMIP3b), following different emission scenarios, and using various GCMs. Data from the previous simulation round (ISIMIP2) have already been used in several studies of historical and future water resources (e.g. Boulange et al., 2021; Gudmundsson et al., 2021; Lange

et al., 2020; Pokhrel et al., 2021; Reinecke et al., 2021).

Based on such climate projections, it is possible to derive meteorological drought indicators, which are determined by precipitation (P) and the atmospheric evaporation demand ($ET_0$). These meteorological droughts propagate into agricultural and hydrological droughts, characterized by a reduction in the soil water content and a reduction in streamflow. Over this century, droughts are expected to become more frequent in the northern hemisphere (Sheffield and Wood, 2008), in most parts

of Europe (Spinoni et al., 2018, Grillakis, 2019), and especially in southern Europe (Pokhrel et al., 2021; Russo et al., 2013, Ruosteenoja et al., 2018). Common meteorological drought indices are directly associated to variations in P and $ET_0$ (Vicente-Serrano et al., 2015). The difference between these two fluxes ($P-ET_0$), also referred to as the climatic water balance, has served as proxy to investigate drying trends (Greve et al., 2014; Prăvălie et al., 2019). For agriculture, $P-ET_0$ is also a major factor determining the need for additional water, i.e. for irrigation.

In Europe, rainfall fulfills the largest part of the crop water requirement (green water), but irrigation (blue water) becomes essential in the most southern parts of the continent (Chiarelli et al., 2020; Liu and Yang, 2010; Siebert and Döll, 2010). For the past decades, the yearly net irrigation requirement in Europe has been estimated between 53 to 1120 mm year$^{-1}$ in Denmark and Spain, respectively (Wriedt et al., 2009). The effectively applied amounts of irrigation could be much lower or higher but are unknown due to the lack of good observational data (Massari et al., 2021). Future global and regional irrigation trend

assessments have commonly used hydrological models (e.g. WaterGAP [Döll and Siebert, 2002] in Döll, 2002; Schadlach et al., 2012), agro-ecosystems models (Lund-Potsdam-Jena managed Land model [LPJmL, Bondeau et al., 2007] in Fader et al.,

2016; Konzmann et al., 2013), or agro-ecological zone (AEZ) models (FAO-AEZ methodology applied in Fischer et al., 2007). The earliest global study addressing future irrigation requirement under climate change was performed by Döll (2002) using the WaterGAP model for two GCMs. The results indicate clear effects on the long-term average irrigation requirement, with

an average global increase of ~10% by the 2070s under the IPCC IS92a scenario (Leggett et al., 1992). Similar increases were found later by Fischer et al. (2007) also using two GCMs applied to an emission scenario from the IPCC Special Report on Emission Scenarios (SRES A2r; Nakicenovic et al., 2000; Riahi et al., 2007). By contrast, global decrease in irrigation water demand were simulated by Pfister et al. (2011) and Konzmann et al. (2013) for the end of the century. These studies only assessed one emission scenario, both from the IPCC SRES (Nakicenovic et al., 2000), namely the A1B and A2 scenario,

respectively. However, in Europe, all these studies indicate clear increases in irrigation water requirement for most parts of the continent where irrigation is currently applied.

The outcomes of the different irrigation assessments can diverge quite significantly. Wada et al. (2013) provided an ensemble of seven General Hydrological Models (GHMs, including LPJmL and WaterGAP) and analyzed the sources of uncertainty on the final predictions. The results showed that the fraction of the variance due to the GCMs is larger than the fraction caused

by the future emission scenarios, and that more than 50% of the variance resulted from the GHMs. The experiment setup also plays a major role as many parameters can influence the irrigation requirement. Global figures are highly different depending on whether the expansion of irrigated areas is considered or not, which explains why Fischer et al. (2007) found increases in the average water requirement, whereas Pfister et al. (2011) and Konzmann et al. (2013) expected a global decrease. The conclusions also depend on (i) whether irrigation efficiencies are considered (i.e., including socio-economic factors), (ii) the

delineation of the growing season (a whole year, fixed or flexible start) and (iii) the type of implementation of irrigation in the model (gross or net requirement, threshold to trigger irrigation, amount of water applied; Telteu et al., 2021).

The irrigation requirement can also be estimated with crop models, which have the added benefit of estimating future trends in crop production and thereby provide useful information to farmers and decision-makers in their adaptation management strategies under climate change. Crop models mainly aim to present quantitative knowledge about the crop development and

crop yield for a given crop having specific features and subject to given environmental conditions (Monteith, 1996). Crop modelling integrates physiological processes and the interactions between the crop and its environment. Several studies have shown the added value of upscaling field-scale crop models to a regional level (e.g. Balkovič et al., 2013; Boogaard et al., 2013; de Wit and van Diepen, 2007; Stöckle et al., 2014), allowing current and future crop yield and irrigation assessments. Liu and Yang (2010) used a GIS-based version of the EPIC (Williams et al., 1989) crop model to spatially evaluate the crop

consumptive water use, partitioning the precipitation input, and the irrigation requirement for the year 2000. Pfister et al. (2011) used CROPWAT (Smith, 1992) to compute the global increase in irrigation requirement to meet future food and biomass demands. Elliot et al. (2014) provided estimations of the potential irrigation water consumption with 10 GHMs (similarly to Wada et al., 2013) and six Global Gridded Crop Models (GGCMs developed within the Agricultural and Model Intercomparison Project (AgMIP) framework, Rosenzweig et al., 2014), of which three are upscaled site-based crop models.

Global-scale crop modelling remains challenging, especially at coarser resolutions (e.g. 0.5° x 0.5° lat-lon), where one grid

cell may contain information of many heterogeneous agricultural fields (Müller et al., 2017). In addition, field management practices (e.g. irrigation practices, fertilizer application) are even more challenging to integrate at regional and global levels. In this study, the spatial version of AquaCrop developed by de Roos et al. (2021) will be used. AquaCrop (Steduto et al, 2009; Raes et al., 2009) set up as a field-scale model, was developed by the Food and Agriculture Organization of the United Nations

(FAO) and is based on the soil water balance. Compared to other, more complex canopy-level models, AquaCrop stands out by its relatively few and intuitive input parameters (Steduto et al., 2009). AquaCrop has already been used in regional agricultural climate impact studies by Dale et al. (2017), where an open-source version of AquaCrop (AquaCrop-OS; Foster et al., 2017) was used to project crop yields for a high number of GCMs under different climate scenarios at a resolution of 2° x 2°.

In this study, the impact of climate change on the future net irrigation requirement is assessed for different emission scenarios and GCMs, using the spatial version of AquaCrop (de Roos et al., 2021) forced with ISIMIP3 meteorological data over the European continent for the first time. First, the model performance is evaluated by comparing historical spatial AquaCrop v6.1 simulations without any irrigation, forced with (i) reanalysis data from ISIMIP3a and (ii) GCM-based meteorological data from ISIMIP3b, against satellite-based surface soil moisture (SSM) observations. Next, AquaCrop v6.1 simulations are

performed using an ensemble of five ISIMIP3b GCMs as forcing to provide estimates of changes in the net irrigation water requirement ($I_{net}$) during the summer months (June, July, August) for two periods in the future (2031-2060 and 2071-2100). The focus is mainly on estimating water demand during the summer period and not on crop water productivity. The objective is to regionally quantify the mean and interannual variability in summer $I_{net}$ for a near and future climate period, and relate this to the current (baseline) $I_{net}$ and future changes in P-$ET_0$ following various climate scenarios. Compared to previous studies,

the advantages are that the simulations are performed with (i) climate data from the latest generation of reanalyses and GCMs, (ii) the most recent set of future scenarios, and (iii) a crop model (AquaCrop), in which the dynamic interactions between water and vegetation are the main focus and where irrigation and management practices can be included with more detail than in a land surface or hydrological model. Future $I_{net}$ projections could be used to inform on climate change adaptation strategies (e.g., climate-smart irrigation, crop type selection, water conservation). The new AquaCrop-ISIMIP3 model setup can be run

at any spatial domain and resolution, providing future opportunities for further climate analysis, also including other irrigation practices and management options.

## 2 Model and data

### 2.1 Model setup

The study domain focuses on the part of the European continent with latitudes (lat) ranging from 34.75° N to 59.75° N and

longitudes (lon) from -10.75° E to 41.25° E. The spatial and temporal resolutions of the model simulations are set to those of the ISIMIP3 input datasets, i.e. 0.5° lat x 0.5° lon, and daily time steps. The same spatial AquaCrop (v6.1) model structure as described by de Roos et al. (2021) is used for this study, but adaptations are made to the spatial resolution, input datasets, and

simulated periods. Simulations are performed from 1985 through 2100, either with or without considering irrigation, and with the respective associated crop-related parameters.

## 2.2 Model parameters

Climate impact assessments are subject to large uncertainties, which increase with longer temporal projections. Therefore, several assumptions are made in this study to limit the uncertainty of other factors than climate. We will present net irrigation requirement values that are independent of the irrigated area, period, infrastructure and the exact crop type. First, simulations are performed over all pixels of the entire study domain (i.e. the main European continent), and the irrigation estimates for the entire hypothetically irrigated agricultural domain are normalized by area to make the results independent of the actual irrigated area. This avoids the need to include estimates of future hypothetical land use (Prestele at al., 2016), and the uncertain evolution of the extent of irrigated areas (Schaldach et al., 2012; Hurtt et al., 2020). Second, the spatial resolution of this study matches that of the ISIMIP input data resolution. In contrast to fine-scale agricultural studies, usually assessing actual irrigation under historical conditions, future climate projections are dependent on the resolution of the driving climate models (or downscaled output). Such studies mainly aim at estimating the irrigation requirement that is needed for crop root uptake, thereby omitting the part of irrigation that is lost to the atmosphere, or retained on the soil surface or in the soil profile. Also, state-of-the-art global and continental-scale climate impact assessments are typically performed at the same resolution (e.g., Jägermeyr et al., 2021; Lange et al., 2020; Thiery et al., 2021). Third, each pixel is defined as a hypothetical homogeneous field, in which the vegetation conditions are identical. For future projections, the use of a representative field crop is supported by the current lack of detailed year- and location-specific crop maps, and by the unpredictability of changes and developments in crop type and distribution. Finally, the uncertainty and high spatial and temporal variability of the start and end of the growing season (King et al., 2018; Menzel and Fabian, 1999; Schadlach et al., 2012) restricts the modelling possibilities. Some previous studies (e.g., Elliott et al., 2014; Fader et al., 2016; Fischer et al., 2007; Konzmann et al., 2012) have used dynamic growing seasons, but the choice has been made to avoid this additional level of uncertainty for this study. Therefore, only the summer months are considered to make the future requirement directly comparable to the baseline $I_{net}$. On average, these are the months presenting the highest $I_{net}$ (Siebert and Döll, 2010), and are expected to remain important months for irrigation requirements, even if growing seasons might shift in the future.

Soil data is extracted from the ISIMIP3 soil input dataset that has been used in the AgMIP GGCM intercomparison (GGCMI; Rosenzweig et al., 2014). ISIMIP3 uses the Harmonized World Soil Database version 1.2 (HWSD1.2), aggregated to 0.5° resolution. The soil dataset represents dominant soil types on croplands within each pixel. Two soil layers are implemented in AquaCrop: one topsoil layer of 0.30 m, and an underlying layer of 1 m, both with the same ISIMIP (topsoil only) textural properties (clay, sand, silt fractions) and gravel content, but with different derived soil hydraulic parameters. More specifically, the volumetric soil water content at saturation, field capacity, and permanent wilting point ($\theta_s$, $\theta_{FC}$, $\theta_{PWP}$) and the saturated hydraulic conductivity ($K_{sat}$) are derived using depth-specific (topsoil, subsoil) pedotransfer functions described by De Lannoy et al. (2014). Because the crop rooting depth is set to 1 m and various bedrock maps indicate that the soil depth over Europe

reaches below 1 m (Dirmeyer and Oki., 2002; Mahanama et al., 2015; Shangguan et al., 2017), no limitations to root development need to be considered (Raes et al., 2009). A total profile depth of 1.30 m is defined, but without the presence of a groundwater table or confining layers, the actual depth below the maximum rooting depth has no influence on the simulations. For the historical model evaluation with satellite retrievals, the choice is made to use a general C3-type of crop with a 1-m
rooting depth to describe the vegetation component, similar to de Roos et al. (2021). This choice is motivated by the coarse spatial resolution and the high uncertainty in crop modifications over time and follows the methodology of well-known hydrological and land surface models, that also make use of general vegetation descriptions (e.g. Niu et al., 2011; Rodell et al., 2004). C3 crops are dominant in Europe (Monfreda et al., 2008; Still et al., 2003). A detailed description of the crop characteristics is given in Table 1 of de Roos et al. (2021). For the model evaluation, no irrigation is activated, the soil fertility
stress of 30% is maintained (de Roos et al., 2021), and the AquaCrop default record of mean annual $CO_2$ concentration observed at Mauna Loa (Hawaii, USA) is considered in the simulations. By contrast, the simulations with irrigation follow the yearly $CO_2$ concentrations of the emission scenarios from ISIMIP3 and assume near-optimal soil fertility.

For the determination of $I_{net}$, a representative field crop is considered. The crop characteristics that determine crop transpiration and hence $I_{net}$ are listed in Table 1. The considered crop transpiration coefficient of 1.10 is a good indicative value of the basal
crop coefficient for the mid-season for a large range of field crops (Allen et al., 1998). Moreover, it is assumed that in the summer months (in which $I_{net}$ is determined) the crop has reached its maximum canopy cover and is prior to senescence. Since $I_{net}$ is determined by keeping the soil water content in the root zone above 50 % of the readily available water (RAW, which is 25 % of the total available water, TAW, for the representative field crop), water stress does not affect crop transpiration. Also, air temperature stress affecting crop transpiration will be small or absent in the summer months with the settings of the
thresholds in Table 1. To be sure of a well-developed crop canopy during the three summer months, it is assumed in the simulations with irrigation, that the crop germinated in early spring, and that the natural crop senescence occurred in late autumn. Irrigated fields are assumed to be well-managed. Hence, a near-optimal soil fertility is defined in AquaCrop, corresponding to a potential achievable biomass production (without any other stress) of 80% (compared to 70% for the simulations without irrigation). Future elevated $CO_2$ concentrations are expected to increase biomass production by reducing
crop transpiration and stimulating crop production ($CO_2$ fertilization effect; Vanuytrecht et al., 2012). This response can vary according to intrinsic crop characteristics or nutrient availability (Vanuytrecht et al., 2011). To avoid overexpression of this effect, the sink term in AquaCrop is lowered to 0%.

**Table 1.** Characteristics of the representative field crop

| Crop parameters | values | units |
|---|---|---|
| Canopy cover (during the 3 summer months) | 85 | % soil cover |
| Crop transpiration coefficient when canopy is complete | 1.10 | - |
| Maximum effective rooting depth | 1.0 | m |
| Soil water depletion at which stomata starts to close | 50 | %TAW |
| Base temperature, below which crop development does not progress | 8.0 | °C |
| Minimum growing degrees required for full crop transpiration | 10.0 | °C-day |

## 2.3 Meteorological data

The AquaCrop model is run with both reanalysis (ISMIP3a) and GCM-based (ISIMIP3b) meteorological input. The ISIMIP3a forcing data extend up to end 2016 and are based on bias-corrected ECMWF Reanalysis data fifth generation (ERA5; Cucchi et al., 2020; Lange, 2019a). The GCM (ISIMIP3b) data start in 2015 and are derived from five different GCMs contributing to the Coupled Model Intercomparison Project phase 6 (CMIP6): GFDL-ESM4, IPSL-CM6A-LR, MPI-ESM1-2-HR, MRI-ESM2-0, UKESM1-0-LL (Lange, 2019b, 2020). These future climate data are separated into different scenarios, which are based on the new scenario framework described by van Vuuren et al. (2014), combining RCPs (van Vuuren et al., 2011) with pathways of socioeconomic development (shared socioeconomic pathways SSPs; O'Neill et al., 2014). The sixth assessment report of the IPCC (2021) demonstrates its results based on this scenario architecture. Scenarios are referred to as SSP$x$-$y$, where SSP$x$ refers to the SSP (five in total, described in O'Neill et al., 2014), and $y$ refers to the level of radiative forcing (in W m$^{-2}$) in 2100 (RCP). Three scenarios are evaluated, SSP1-2.6 (low emissions thanks to strong mitigation), SSP3-7.0 (high emissions), and SSP5-8.5 (extreme emissions or unmitigated), for five GCMs, resulting in a total of 15 SSP-GCM scenarios. Under SSP3-7.0 and SSP5-8.5, a global warming of 2 °C will likely be exceeded by mid-century.

AquaCrop requires minimum and maximum temperature, rainfall, and reference evapotranspiration (ET$_0$), on a daily basis. Meteorological variables extracted from ISIMIP3 are the daily maximum and minimum temperatures, total precipitation, near-surface relative humidity, near-surface wind speed (at a 10 m height), and the shortwave downwelling radiation. Daily ET$_0$ values are estimated with the FAO Penman-Monteith equation according to the guidelines presented in the FAO Irrigation and Drainage Paper 56 (Allen et al., 1998) with the available variables and ISIMIP elevation data (for the estimation of the atmospheric pressure).

## 2.4 Satellite-based evaluation data

To evaluate the performance of the regional AquaCrop simulations forced with ISIMIP input, two L-band microwave-based level 2 SSM products are used: (i) the SMUDP2 data product version 650 from the ESA Soil Moisture Ocean Salinity (SMOS)

mission (Kerr et al., 2010), from 2011 onwards; and (ii) the SPL2SMP product version 7 from the NASA Soil Moisture Active Passive (SMAP) mission (Chan et al., 2016), from 2015 onwards. For both data sources, only recommended quality retrievals are included. Additionally, retrievals for daily minimum temperatures below 4 °C are screened out to avoid retrievals near
frozen conditions. Both satellite products are projected on a 36-km Equal-Area Scalable Earth version 2 (EASEv2) grid, for SMOS data after reprojection as in De Lannoy and Reichle (2016). It should be noted that SMOS data over Europe have been affected by radio frequency interference and are filtered out, especially in the early years after launch in 2010 (Oliva et al., 2012).

## 3 Methodology

**3.1 Simulations**

Three types of simulation experiments are performed and referred to as SIM1, SIM2, and SIM3, with the corresponding settings described in Table 2. SIM1 and SIM2 constitute the historical model evaluation against satellite retrieval products. For SIM1, reanalysis meteorological data (ISIMIP3a) are used as input and simulated SSM is compared to satellite reference data at a daily resolution (short-term variability). AquaCrop is run over the study area for the period from 1 January 2011 through 31
December 2016 with reanalysis data, i.e. until the end of the available reanalysis data. The second set of historical simulations (SIM2) are GCM-driven (ISIMIP3b) SSM simulations. The purpose of SIM2 is to determine whether the GCM-based forcing is reliable to use for future simulations, i.e. via an evaluation of multi-year average SSM (long-term distribution). For each GCM, AquaCrop is run with climate input data for the period 2011-2020. These input data gather historical simulated climate for 2011-2014, and scenario-based simulated climate for the period 2015-2020, only accounting for SSP5-8.5 (only small
differences occur between the three SSPs for this time period). The meteorological time series of the two periods are stitched together to provide continuous AquaCrop forcing fields for 2011-2020. SIM1 and SIM2 have a spin up period of four years, and only output from 2015 onwards is used for evaluation, i.e. starting when both SMOS and SMAP data are available.

Once the model has been evaluated with the first two experiments, simulations of SIM3 are run with GCM-driven meteorological input (ISIMIP3b) for the baseline (historical reference period, 1985-2014) and into the future from 2021
through 2100. Irrigation is activated in AquaCrop and the net irrigation water requirement $I_{net}$ for the three summer months is extracted from the simulations for the reference time window, and two future time horizons (near future 2031-2060, and far future 2071-2100). For the baseline simulation, the initial moisture conditions are set to field capacity while the future periods have a spin up of at least 10 years (continuous simulation from 2021 through 2100). For SIM3, irrigation is introduced, using the net irrigation requirement option in AquaCrop, whereby a small amount of water (just covering the crop ET for that day)
is injected into the root system on days when a certain fraction of the RAW is depleted (Raes et al., 2017). With this option, solely the amount water taken up by the roots is considered, where the wetting of the soil surface, and interval and application amount specific to a particular irrigation method are not relevant. By selecting a threshold of 50% RAW depletion, which is the average depletion in an optimal irrigation interval (Smith, 1992), crop water stress affecting the canopy development and

transpiration of the representative field crop is avoided, and effective rainfall (the part stored in the root system up to field

capacity) is still considered. All simulations performed in this research are uncoupled, i.e. feedback mechanisms from irrigation

on atmospheric climate (e.g. Hirsch et al., 2017; Thiery et al., 2017, 2020) are neglected.

**Table 2** Description of the different simulations with regard to the simulation periods, analyses, climate data, crop characteristics, soil fertility stress and the activation of the irrigation (ON/OFF).

| Sim. | Period | Analysis | Climate data | Crop | Soil fertility stress (%) | Irrigation |
|------|--------|----------|--------------|------|---------------------------|------------|
| **SIM1** | April 2015 - December 2016 | SSM short-term evaluation | reanalysis | generic C3 | 30 | OFF |
| **SIM2** | April 2015 - December 2020 | SSM climatological evaluation | 5 GCMs | generic C3 | 30 | OFF |
| **SIM3** | 1985-2014 2031-2060 2071-2100 | I$_{net}$ projections | 5 GCMs 5 GCMs x 3 SSPs | representative field crop | 20 | ON |

## 3.2 Evaluation of historical AquaCrop simulations

### 3.1.1 Skill metrics

To compare the spatial and temporal patterns of SSM from 0.5° AquaCrop simulations with 36-km satellite data, nearest-neighbour sampling is used to spatially match simulated SSM with SMOS and SMAP retrievals. The output variable extracted from the AquaCrop simulation is the volumetric water content of the topsoil compartment, corresponding to the first 0.1 m of the soil (output variable WC01 in AquaCrop). After quality screening of the satellite data (see section 2.4), about 0.9 and 1.9

million usable observations are kept over the study domain (composed of 3882 pixels) for SMOS and SMAP, within the period April 2015 – December 2020. The most widely used validation metrics for SSM estimates from large-scale model simulations and retrievals are the Pearson correlation coefficient (R), the bias, the root-mean-square difference (RMSD), and the unbiased RMSD (ubRMSD), which are calculated as follows:

$$R = \frac{\sum_{n=1}^{N}(x_n - \bar{x})(y_n - \bar{y})}{\sqrt{\left(\sum_{n=1}^{N}(x_n - \bar{x})^2\right)\left(\sum_{n=1}^{N}(y_n - \bar{y})^2\right)}} \quad (1)$$

$$bias = \frac{1}{N}\sum_{n=1}^{N}(x_n - y_n) \quad (2)$$

$$RMSD = \sqrt{\frac{1}{N}\sum_{n=1}^{N}(x_n - y_n)^2} \quad (3)$$

$$ubRMSD = \sqrt{RMSD^2 - bias^2} \quad (4)$$

where x are the simulated SSM, y the reference observations, N the number of observation-simulation pairs, and ($\bar{\phantom{x}}$) is the temporal mean. A minimum threshold of N=100 reference data points in time are set per pixel for all analyses. Anomaly

correlations are discussed in Appendix A. The aim of the historical evaluation is to assess the performance of AquaCrop to integrate ISIMIP3 meteorological forcings and to provide SSM estimates. By design, the model and satellite retrievals are biased, due to model parameters related to the soil and the uniform vegetation type (generic C3 crop), vertical representativeness bias, etc. Therefore, bias-free metrics (R and ubRMSD) are essential to assess whether the main temporal variations of SSM are captured by the model forced with ISIMIP3 data.

### 3.1.2 Difference in evaluation for SIM1 and SIM2


Both the time series of historical SIM1 and SIM2 SSM are compared to satellite observations through the skill metrics described in section 3.1.1 for the time period with available data for both SMOS and SMAP, i.e. from April 2015 through 2016. SIM1 is a short-term evaluation since daily SSM simulations are compared to satellite observations. All months of the year with available and qualitative satellite data were included in this first validation step.

For the historical SIM2 SSM simulations, the multi-year average (long-term) results driven by the five different GCMs, and the median SSM time series across the GCMs are evaluated. However, for each simulation year, only the period between the $1^{st}$ of March up to the $31^{st}$ of October is considered, because only summer months will be considered for the subsequent analysis of future $I_{net}$ (section 3.2). Climate models are developed to indicate changing climatic trends but do not present daily accurate data, if they are not constrained by observational data. Therefore, the multi-year average (i.e. climatology) of SIM2 SSM is

computed and then be compared to the climatologies of satellite SSM during the observation period, using the same skill metrics presented in section 3.1.1. Climatologies are calculated using a sliding window of 31 days with a minimum threshold of three data points of data within the window. The computation of the climatology is restrained to the availability of reference satellite data (i.e. SMAP, data available from April 2015), as it is also the case in satellite data assimilation systems (e.g., SMAP Level 4 product; Reichle et al., 2019).

### 3.2 Future net irrigation $I_{net}$ requirement (SIM3)


This study focuses on the evaluation of the change in $I_{net}$ during the period for which the highest irrigation demand is expected in all parts of Europe, i.e. June, July, and August (Siebert and Döll, 2010). For the evaluation of the future irrigation water requirement, daily $I_{net}$ values (directly available from the model output) are first extracted from the SIM3 output of the 15 different SSP-GCM combinations. The results are expressed in mm month$^{-1}$ by averaging the $I_{net}$ of the three summer months.

The summer irrigation is then used for evaluation following two approaches. First, the summer $I_{net}$ is averaged over the 30-year time window allowing to compare future (2031-2060 and 2071-2100) and baseline (1985-2014) average $I_{net}$ by computing the difference ($\Delta I_{net}$). A statistical t-test is carried out to define whether the difference of mean $I_{net}$ between the two periods is significant ($p < 0.05$). Second, interannual variation is assessed based on the $I_{net}$ range ($RI_{net}$), defined as the difference between maximum and minimum summer $I_{net}$ of the 30-year time window. Again the difference between future and baseline $RI_{net}$ is

evaluated ($\Delta RI_{net}$). $I_{net}$ simulated for the different SSP-GCM combinations are analyzed individually. Additionally, the median

results across the GCMs for each scenario are presented. A simple climate index (P-ET$_0$), computed for the three summer months, is used to identify where drying trends are potentially occurring, and how this is reflected in the irrigation requirement.

## 4 Results

### 4.1 Evaluation of historical regional AquaCrop simulations forced with ISIMIP3

#### 4.1.1 Short-term evaluation (SIM1)

AquaCrop SSM simulations forced with ISIMIP3a reanalysis data are evaluated with SMOS and SMAP SSM retrievals from April 2015 through December 2016 (start when data from both missions are available, until end of reanalysis data). The spatially averaged skill metrics for AquaCrop SSM compared to satellite observations from SMOS and SMAP are presented in Table 3. The skill is generally better relative to SMAP SSM than relative to SMOS SSM. The expected errors of both

missions are 0.04 m³m$^{-3}$ when comparing the satellite data to in situ reference data (Entekhabi et al., 2014). Here, slightly higher ubRMSDs of 0.06 and 0.08 m³m$^{-3}$ are obtained.

**Table 3** Spatial mean (± spatial standard deviation) of R, RMSD, bias, and ubRMSD between SIM1 SSM estimates, SMOS, and SMAP, for April 2015 through December 2016.

| Reference obs. | R (-) | RMSD (m³m$^{-3}$) | bias (m³m$^{-3}$) | ubRMSD (m³m$^{-3}$) |
|---|---|---|---|---|
| **SMOS** | 0.53 (±0.13) | 0.10 (±0.03) | -0.05 (±0.05) | 0.08 (±0.02) |
| **SMAP** | 0.65 (±0.15) | 0.08 (±0.04) | -0.03 (±0.06) | 0.06 (±0.01) |

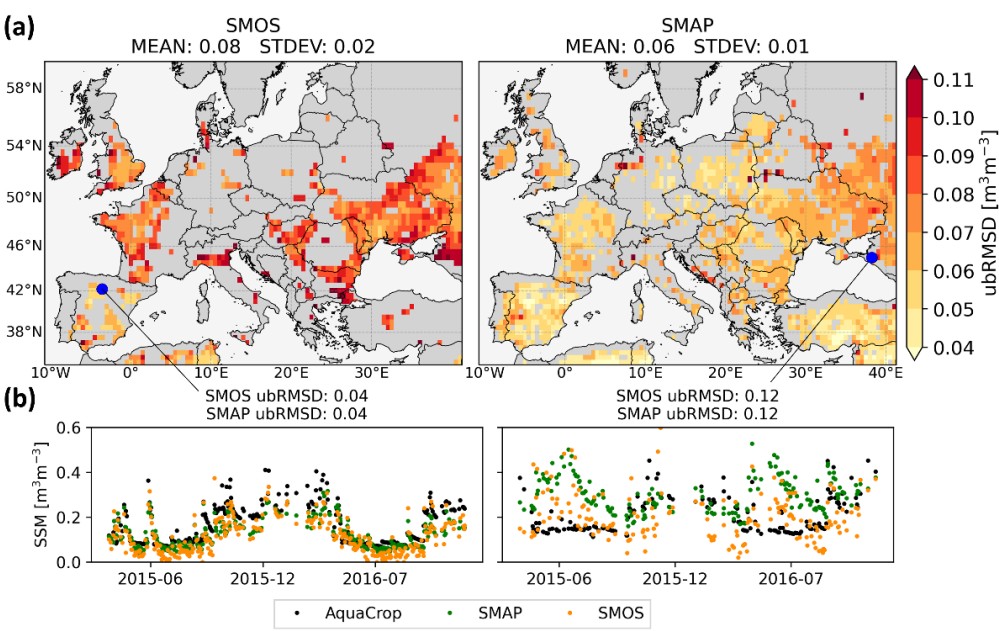


**Figure 1** (a) ubRMSD of SIM1 AquaCrop SSM compared to SMOS (left) and SMAP (right) retrievals for April 2015 through December 2016. The spatial mean and standard deviation are indicated in the titles (MEAN, STDEV). Grey areas correspond to pixels where the number of satellite retrievals is smaller than 100. (b) SSM time series of two pixels (marked by blue dots in (a) with the title indicating the ubRMSD [$m^3m^{-3}$] against SMOS and SMAP for each location.

The spatial distribution of ubRMSD is presented in Fig. 1a. For further discussion, a partitioning of the study domain in various zones is shown in Appendix A (Fig. 1A). Simulated SSM deviate more from SMOS retrievals in north and central-eastern Europe, whereas pixels located in southern regions (e.g. Spain) present a better model performance when comparing to SMOS. Central-eastern Europe presents on average a higher ubRMSD, stressing a lower performance in this region. Time series of SSM estimates at two locations are shown in Fig. 1b. The modelled SSM contents are close to satellite retrievals for the first

pixel (left), and a mismatch is found between simulations and retrievals for the second pixel (right). For the latter, AquaCrop simulations are underestimating SSM during summer and it can be noticed that SMOS and SMAP retrievals substantially diverge for this location.

### 4.1.2 Climatological evaluation (SIM2)

The SIM2 AquaCrop SSM for the period 2011-2020 is forced with ISIMIP3b GCM-driven meteorology. The modelled SSM

is converted to a multi-year average climatology for the five GCMs, and compared to climatologies of SMOS and SMAP SSM (2015-2020) for the months March through October. Spatially averaged temporal skill metrics are shown in Fig. 2.

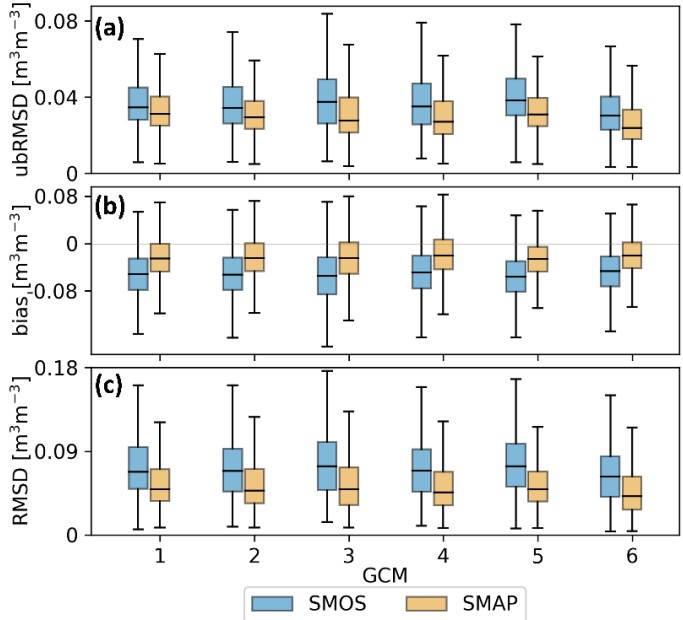

**Figure 2** Spatial boxplots of (a) ubRMSD, (b) bias, and (c) RMSD, for five GCMs (1 to 5: GFDL-ESM4, IPSL-CM6A-LR, MPI-ESM1-2-HR, MRI-ESM2-0, UKESM1-0-LL) and the median across GCMs (6). SIM2 SSM is compared with SMOS (blue, left) and SMAP (yellow-

brown, right) SSM for April 2015 through December 2020. Only SSM values for the months March through October are considered in the computation of the skill metrics, and the spatial coverage is different for SMOS and SMAP. The boxes represent the values in the interquartile

range (IQR), the line in the box corresponds to the median, and the whiskers extend to Q1 – 1.5IQR and Q3 + 1.5IQR, or are cut off if all data points fall into the interval (outliers are not shown).

All GCM-driven simulations are similarly biased compared to the satellite products. The larger dry bias with SMOS (on average -0.05 m³m⁻³) compared to SMAP observations (on average -0.02 m³m⁻³) agrees with the short-term evaluation results of the reanalysis-driven simulations (section 4.1.1). The evaluation of predicted SSM compared to satellite data results in spatially averaged mean ubRMSDs ranging between 0.02 and 0.04 m³m⁻³, with the lowest values for the multi-model median SSM (Fig. 2a).

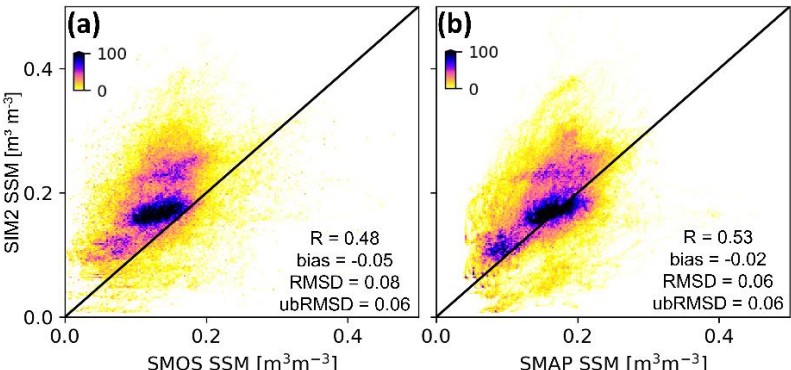

**Figure 3** Density scatter plots comparing the SIM2 SSM climatology (median climatology across the GCMs), against a reference climatology based on (a) SMOS retrievals, (b) SMAP retrievals (April 2015-2020). Only the period from 1 March to 31 October is considered. The colorbars represent the number of space-time samples per bin. Spatio-temporal skill metrics (R, bias, RMSD and ubRMSD) are shown.

Figure 3 presents the spatio-temporal skill metrics comparing the multi-model median SIM2 SSM climatology with the two satellite-based SSM climatologies. The GCM-driven SSM climatology remains close to satellite SSM climatologies in drier conditions but there is a wet model bias (or dry satellite retrieval bias) in wetter conditions (Fig. 3). Correlations between simulated climatologies and satellite data are slightly lower when considering individual GCMs (no median) with ranges of 0.41-0.45 and 0.47-0.51 for SMOS and SMAP, respectively (not shown). By design, GCM climatolgies are unbiased against the reanalysis climatology, indicating that GCM-driven projections are representative of the reanalysis climate. From the evaluation of SIM1 and SIM2, it can overall be concluded that AquaCrop forced by ISIMIP3 input demonstrates a reasonable performance in terms of spatio-temporal SSM pattern representation; we therefore assume that the model can be used to project $I_{net}$ changes across the study area.

## 4.2 Future net irrigation requirement $I_{net}$ (SIM3)

### 4.2.1 Climate impact on mean $I_{net}$

The change in summer $I_{net}$ is assessed by the difference ($\Delta I_{net}$) between the mean $I_{net}$ of the future horizons (2031-2060; 2071-2100) and the baseline period (1985-2014). In Fig. 4, spatial boxplots of $\Delta I_{net}$ are presented for five GCMs individually and for the median across the GCMs for each scenario.

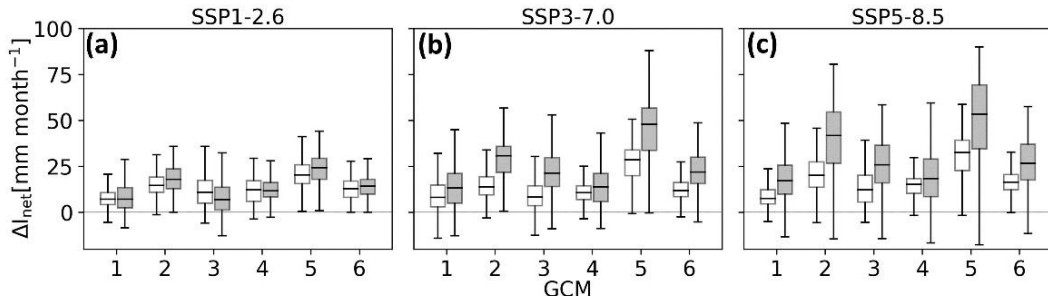

**Figure 4** Spatial boxplots of $\Delta I_{net}$ [mm month$^{-1}$] for the three SSPs (a, b, c) and five GCMs (1 to 5: GFDL-ESM4, IPSL-CM6A-LR, MPI-ESM1-2-HR, MRI-ESM2-0, UKESM1-0-LL) and the median across GCMs (6), for near (2031-2060: white) and far (2071-2100: grey) future, both relative to the baseline period (1985-2014).

Based on Fig. 4, increases in $I_{net}$ are expected in the future for all scenarios, where the severity of the increase depends on the emission scenario. SSP1-2.6 presents a stabilization of $I_{net}$ towards the end of the century in line with the evolution of $CO_2$ for this scenario, whereas the other scenarios show increases from 2031-2060 to 2071-2100. The differences between the GCMs within an SSP are considerable and these disparities increase with rising emission scenarios. According to the first GCM (GFDL-ESM4), on average about 7 mm month$^{-1}$ extra irrigation water will be required in the summer months by 2050 for SSP5-8.5, whereas for UKESM1-0-LL, more than 20 mm month$^{-1}$ will be required by mid-century for the same emission scenario. Decreases in $I_{net}$ (boxplot whiskers below 0, Fig. 4c) are only observed in a few southern coastal locations under the high and severe emission scenarios. In these historically warm and dry regions with insignificant rainfall in the summer months, the effect of stomatal closure by 5% as response to $CO_2$ concentrations above 550 ppm is stronger than the increase in ET (less than 5%). These negative differences are statistically non-significant (except for GFDL-ESM4, but the total area subjected to decreases is negligible). Figure 5 presents the spatial distribution of $\Delta I_{net}$, for the median across the GCMs. Regions where all GCMs present significant changes are stippled. Once the results are presented in terms of medians, no statistically significant decrease in $\Delta I_{net}$ is observed.

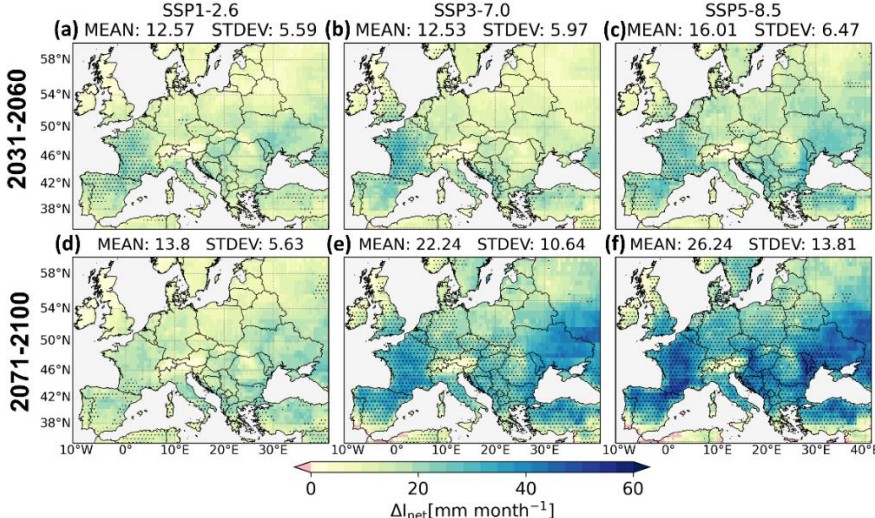

**Figure 5** Changes in summer $I_{net}$ ($\Delta I_{net}$) [mm month$^{-1}$], median across five GCMs for the two future time horizons (rows) and the three scenarios (columns) with reference to the baseline period. The stippled areas represent pixels where all five GCMs present statistically significant changes (t-test, $p < 0.05$).

Under the low emission scenario (Fig. 5a and d), the whole continent will face a mild increase in summer $I_{net}$ by about 13 mm month$^{-1}$ (+18%) in the near and far future, and regions undergoing severe increases cannot be identified. Towards the end of the century, for high and extreme emissions, the most affected areas (where all GCMs agree on a significant change) are situated in the central and southern latitudes (Fig. 5e and f). For the end of the century, the spatial mean summer $I_{net}$ increases by 22 and 26 mm month$^{-1}$ (+30% and +35%) for SSP3-7.0 and SSP5-8.5, respectively. The most eastern parts are on average presenting large $\Delta I_{net}$ for the far future (2071-2100), but according to GFDL-ESM4 alone (not shown), these changes are non-significant and therefore not stippled in Fig. 5e and f. All SSP-GCM combinations agree on the evolution of $I_{net}$ in the northern Alps, where the situation is likely to remain stable, in terms of amounts of required irrigation water.

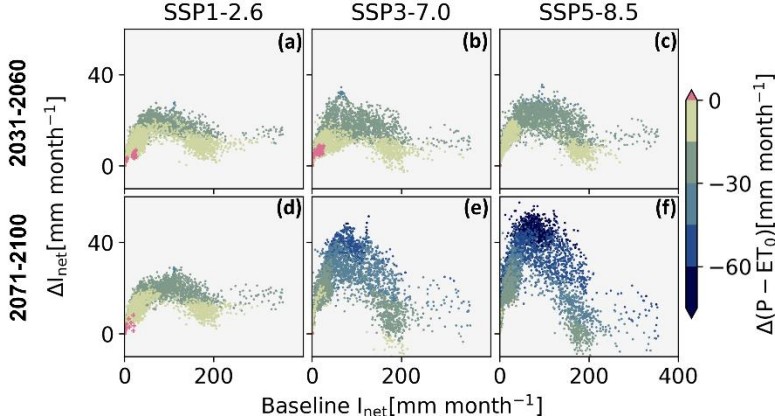

**Figure 6** Scatter plots of $\Delta I_{net}$ relative to the baseline period for the two future periods (rows) and the three scenarios (columns). The coloring refers to the corresponding $\Delta(P\text{-}ET_0)$. Increases in $\Delta(P\text{-}ET_0)$ are represented by pink crosses. All values ($I_{net}$ and $\Delta(P\text{-}ET_0)$) are medians across the GCMs [mm month$^{-1}$].

Figure 6 shows the spatial relationship between the expected change in summer $I_{net}$ with reference to the baseline period. Areas with historically extreme (> 150 mm month$^{-1}$) or low (< 20 mm month$^{-1}$) $I_{net}$ will not see their future needs increase drastically, whereas regions with a relatively moderate to high baseline $I_{net}$ will face the strongest changes. Table 4 summarizes the baseline summer $I_{net}$ and $\Delta I_{net}$ (median and standard deviation across GCMs), for six selected countries and the Benelux included in this study area. The difference between the $\Delta I_{net}$ for various scenarios is of the same order of magnitude as, and often smaller

than, the variability introduced by the various GCMs. Note again that the presented numbers are expressed in mm month$^{-1}$, but only averaged over three summer months, and the results are purely based on climate projections that are integrated into AquaCrop, assuming a hypothetical C3 crop, near-optimal fertilization, and without accounting for the presence or quality of the irrigation network.

**Table 4** Median across GCMs (± standard deviation of GCMs) of baseline summer $I_{net}$, $\Delta I_{net}$, baseline $RI_{net}$, and $\Delta RI_{net}$ [mm month$^{-1}$], spatially averaged over the country, for six European countries and the Benelux. The changes $\Delta$ are presented for the two future horizons (2031-2060 and 2071-2100, columns), and for the different emission scenarios (line 1, 2, 3 of a cell corresponding to SSP1-2.6, SSP3-7.0, and SSP5-8.5, respectively).

| Country | Summer $I_{net}$ (mm month$^{-1}$) | | | $RI_{net}$ (mm month$^{-1}$) | | |
|---|---|---|---|---|---|---|
| | Baseline | Δ 2031-2060 | Δ 2071-2100 | Baseline | Δ 2031-2060 | Δ 2071-2100 |
| **Benelux** | 26 (±2) | 12 (±4) | 14 (±9) | 66 (±14) | 13 (±7) | 8 (±11) |
| | | 14 (±8) | 24 (±14) | | 14 (±16) | 35 (±11) |
| | | 18 (±6) | 31 (±13) | | 9 (±3) | 21 (±11) |
| **France** | 53 (±2) | 19 (±5) | 17 (±6) | 87 (±10) | 6 (±9) | -3 (±8) |
| | | 22 (±6) | 31 (±14) | | 3 (±12) | 10 (±13) |
| | | 24 (±9) | 40 (±12) | | 12 (±13) | 6 (±23) |
| **Germany** | 27 (±2) | 11 (±6) | 13 (±10) | 63 (±15) | 8 (±8) | 1 (±12) |
| | | 13 (±9) | 20 (±18) | | 9 (±9) | 27 (±13) |
| | | 14 (±10) | 27 (±19) | | 19 (±11) | 21 (±16) |
| **Italy** | 92 (±2) | 13 (±3) | 15 (±3) | 77 (±8) | -5 (±12) | -12 (±11) |
| | | 13 (±3) | 18 (±7) | | 3 (±9) | -5 (±11) |
| | | 13 (±6) | 22 (±7) | | 1 (±9) | -4 (±10) |
| **Romania** | 51 (±2) | 16 (±8) | 14 (±7) | 81 (±17) | 7 (±14) | 20 (±9) |
| | | 12 (±11) | 28 (±16) | | 14 (±21) | 8 (±11) |
| | | 19 (±10) | 36 (±14) | | 20 (±10) | 18 (±16) |
| **Spain** | 137 (±1) | 16 (±4) | 13 (±5) | 86 (±5) | -4 (±5) | -6 (±11) |
| | | 17 (±8) | 24 (±11) | | -3 (±8) | -12 (±10) |
| | | 19 (±7) | 28 (±8) | | 0 (±8) | -9 (±5) |
| **Ukraine** | 66 (±3) | 20 (±7) | 20 (±8) | 100 (±10) | -4 (±14) | 9 (±9) |
| | | 13 (±11) | 37 (±17) | | -9 (±14) | 8 (±14) |
| | | 20 (±10) | 41 (±17) | | 14 (±12) | 0 (±11) |

Figure 6 also shows how the atmospheric conditions in the summer, i.e. $\Delta(P\text{-}ET_0)$, are directly related to $\Delta I_{net}$. The largest increases in $\Delta I_{net}$ correlate with strong decreases in P-ET$_0$. The few locations showing a positive $\Delta(P\text{-}ET_0)$ (black crosses in Fig. 6a, b and d) are still subjected to a slight increase in irrigation requirement. The $\Delta I_{net}$ estimates obtained with AquaCrop provide additional information over the mere $\Delta(P\text{-}ET_0)$ estimates, because the soil-plant system has a memory and temporally integrates the past P-ET$_0$ and irrigation events. Since the crop and management parameters are constant for the entire study domain, the only factor affecting $I_{net}$ for a given climate (P and ET$_0$) is the buffering capacity of the root zone, i.e. soil

characteristics. An analysis of the influence of soil characteristics showed, for instance, that sandy soils see their $I_{net}$ enlarge more rapidly compared to loamy soils. However, no clear conclusions could be drawn, because the vast majority of Europe at the resolution of this study is dominated by a loamy soil texture.

### 4.2.2 Climate impact on the interannual variability of $I_{net}$ ($RI_{net}$)

To assess the potential change in interannual variability of summer $I_{net}$, the difference between the maximum and minimum summer $I_{net}$ within a 30-year time period (range of $I_{net}$ = $RI_{net}$) is evaluated. The future $RI_{net}$ values are assessed with reference to the baseline period, resulting in $\Delta RI_{net}$ for each scenario and GCM. Results are presented in Fig. 7, where expansions of $RI_{net}$ are indicated in red, and reductions in blue.

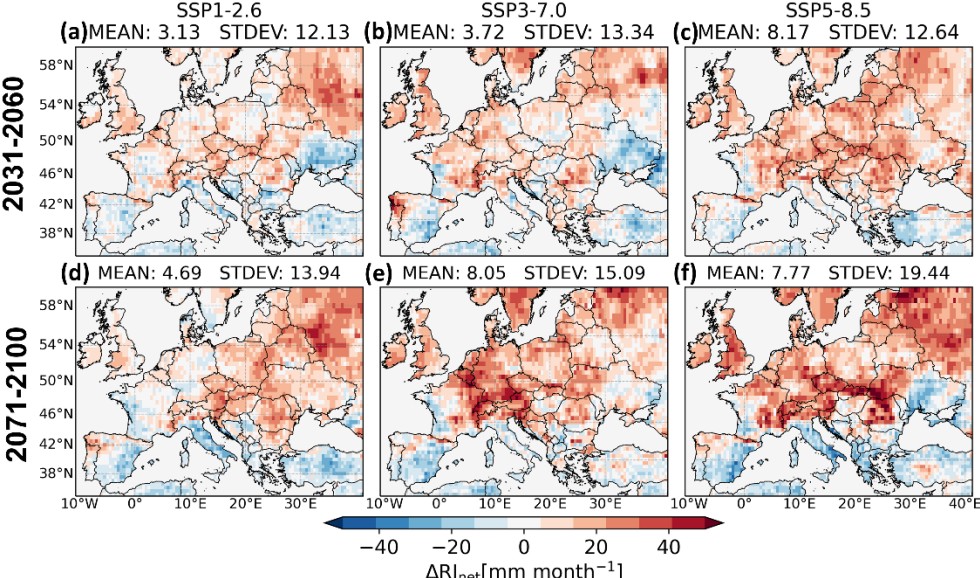

**Figure 7** Future changes in $RI_{net}$ ($\Delta RI_{net}$) [mm month$^{-1}$] median of five GCMs for the two future horizons (rows) and the three scenarios (columns) with reference to the baseline period.

For all SSPs, future $RI_{net}$ are likely to decrease in most of southern Europe, whereas the gap between the highest and lowest irrigation requirement in the 30-year time window is expected to grow in northern and central regions of Europe. Similar to $\Delta I_{net}$ (Fig. 5), Fig. 7 shows that changes are strengthened from SSP3-7.0 to SSP5-8.5 (far future, Fig. 7e and f), in line with the expected increase in extreme events with climate change. Table 4 summarizes the baseline $RI_{net}$ and changes in interannual variability for some selected countries in Europe.

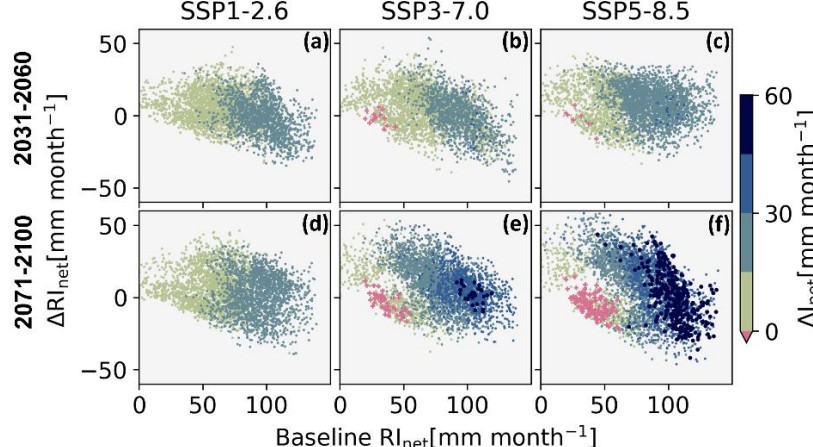

**Figure 8** Scatter plots of $\Delta RI_{net}$ versus baseline range for the two future periods (rows) and the three scenarios (columns). The dots are colored by $\Delta I_{net}$ of the corresponding time window and scenario. Negative $\Delta I_{net}$ are marked with pink crosses and extreme increases in $\Delta I_{net}$ (> 45 mm month[-1]) are represented as dark blue larger dots. All values ($RI_{net}$ and $\Delta I_{net}$) are medians across the GCMs [mm month[-1]].

Figure 8 presents how the change in interannual variability ($\Delta RI_{net}$) of the two future periods relates to the baseline $RI_{net}$, and to $\Delta I_{net}$. Regions with severe increases in $I_{net}$ do not necessarily present the highest enlargements in $RI_{net}$. The largest baseline $RI_{net}$ correlate to lower $\Delta RI_{net}$ for the far future (SSP3-7.0 and SSP5-8.5, Fig. 8e and f), in combination with high values of $\Delta I_{net}$ (dark blue dots, Fig. 8e and f). In other words, the Mediterranean region, west France and the region around Black Sea, with currently a high interannual variability in irrigation requirements will see their requirement significantly increase to more steady high irrigation requirement. Large $\Delta RI_{net}$ values follow the Carpathian Mountains (central Europe) for SSP5-8.5 (Fig. 7f). According to the model, only little irrigation was required in these mountainous regions during the baseline whereas future requirement are projected to increase. In the future, $I_{net}$ peaks to larger values for several years, increasing $RI_{net}$.

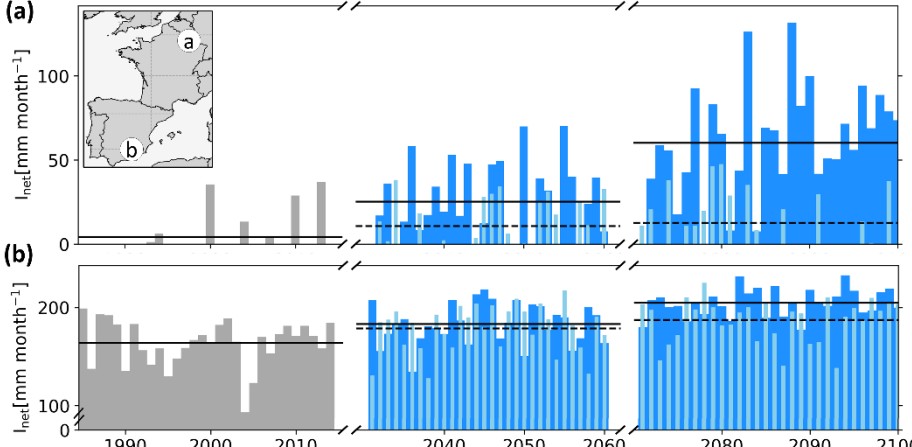

**Figure 9** Time series of summer $I_{net}$ [mm month[-1]] simulated with climate data extracted from IPSL-CM6A-LR for two locations (a) 49.75° N 5.25° E, and (b) 38.25° N 2.75° W, marked on the inset. SSP1-2.6 is represented by thin light blue bars, SSP5-8.5 by dark blue wider bars. The horizontal lines correspond to the time series mean over the climate window, for the future horizons the dotted lines correspond to SSP1-2.6 and the full lines to SSP5-8.5.

To get a better understanding of changes in interannual variability of $I_{net}$, time series at two different locations for one GCM are presented in Fig. 9. Fig. 9a shows the evolution of summer $I_{net}$ for a pixel in central-western Europe, with a $\Delta RI_{net}$ of 100 mm month$^{-1}$ (for IPSL-CM6A-LR, randomly chosen GCM). During the baseline period, summer $I_{net}$ fluctuates between zero and about 35 mm month$^{-1}$, while at the end of the century, the maximum $I_{net}$ of the time window will reach 135 mm month$^{-1}$ for SSP5-8.5 with the same minimum $I_{net}$ as for the baseline. For the second pixel in southern Europe (Fig. 9b), a stabilization of the yearly summer requirement is expected. Overall, more water will be required here, but summer $I_{net}$ will not vary importantly relative to the average requirement from one year to another. This second location results in a decrease in $RI_{net}$ of about 55 mm month$^{-1}$ for the presented GCM under SSP5-8.5.

## 5 Discussion

### 5.1 A new model setup for climate change impact assessment

The regional setup of the AquaCrop model using ISIMIP3 meteorological data has potential to assess impacts of climate change on the irrigation requirement and possibly also on future crop production. First, the short-term evaluation proved that the model forced with reanalysis meteorology (ISIMIP3a) has an acceptable performance, i.e. the ubRMSD between SIM1 SSM simulations and satellite retrievals is 0.06 and 0.08 m³m$^{-3}$, for SMAP and SMOS, respectively (Table 3). The lower model performance compared to SMOS SSM could be due to remaining radio frequency interference contamination (Oliva et al., 2012). It is important to note that the satellite target uncertainty is 0.04 m³m$^{-3}$ over areas with less than 5 kg m$^{-2}$ vegetation water (i.e. excluding dense vegetation; Entekhabi et al., 2014). Even though a conservative screening was used, this target may be exceeded at some times and locations. The findings of the evaluation against SMAP SSM are comparable to the results found by de Roos et al. (2021). However, the latter study used Modern-Era Restrospective analysis for Research and Applications (version 2) input and showed a slightly higher performance between the simulated SSM of the regional AquaCrop model and SMAP SSM. Furthermore, the difference in study domain and especially in resolution play a major role in explaining this difference (larger domain and soil characteristics aggregated to coarser pixels in this study).

The strong agreement between the SIM2 SSM climatologies obtained with GCM-driven input (ISIMIP3b) and the reference satellite SSM climatologies (Fig. 2), further confirmed that the historical GCM-driven input is also reliable. A larger bias was observed in wetter moisture conditions (Fig. 3), possibly coming from the model itself or from biases in satellite retrievals. Overall, the provided simulated atmospheric data could represent the main variations of SSM for the past (2015-2020) and is therefore reliable to be used for climate change assessments.

The model simulations for the historical evaluation did not include any irrigation and therefore, some mismatches in SSM could possibly be expected in areas which are currently irrigated. Earlier studies suggest that the contrast between satellite observations and model simulation could identify unmodelled processes (Brocca et al., 2018). In a separate analysis (not shown), this effect was assessed by evaluating the correlation values with regard to irrigation areas that are equipped for irrigation (AEI; similar to de Roos et al., 2021). By using the FAO global maps of irrigated areas version 5 (Siebert et al.,

2013) aggregated to ISIMIP resolution, pixels were divided into two groups: (1) less than 10% of the area is equipped for irrigation, and (2) more than 10% of the pixel is equipped. The correlations between AquaCrop SIM1 SSM and satellite SSM for the irrigated pixels (AEI>10%) were nearly identical to correlations for the locations with an AEI<10% (0.65 versus 0.67 for SMOS, and 0.51 for both classes of AEI for SMAP), therefore not revealing where irrigation was missing in the simulations.

Even if irrigation could be captured by observation-based SSM (Kim et al., 2020), the low amount of reference satellite data in regions presenting high percentages of AEI compromised the evaluation in this study. The use of a conservative screening of satellite SSM retrievals for both SMOS and SMAP resulted in a significant amount of data loss, especially in densely vegetated areas which are generally masked out (Kim et al., 2020). Further, irrigated areas are usually much smaller than the considered 0.5° pixel and a recent study stressed the low potential to detect irrigation at coarse resolutions (Dari et al., 2021).

Nevertheless, the second time series presented in Fig. 1b showed an underestimation of SSM during summer by the model and large differences between SMOS and SMAP, both suggesting potential irrigation applications, as confirmed by a high AEI percentage.

## 5.2 Future mean and interannual variability of summer $I_{net}$

The evolution of future summer $I_{net}$ with climate change is highly dependent on the scenario (SSP), but also on the GCM

(Table 4, Fig. 4). Our results agree with several earlier droughts and irrigation projection assessments (Döll, 2002; Elliott et al., 2014; Konzmann et al., 2013; Pfister et al., 2011; Pokhrel et al., 2021; Ruosteenoja et al., 2018; Satoh et al., 2021; Schaldach et al., 2012; Wada et al., 2013). Under high and extreme emission scenarios, the whole continent will be significantly impacted by the end of the century (Fig. 5e and f), with the most drastic changes in central and upper southern latitudes of the study domain, confirmed by the high increases in meteorological (Spinoni et al., 2018) and soil water (Ruosteenoja et al., 2018)

shortages in these regions. $\Delta I_{net}$ spatial patterns (Fig. 5) are comparable to the findings of Konzmann et al. (2013) and Wada et al. (2013) for the most affected areas where all GCMs present significant changes. Eastern Europe shows on average large positive $\Delta I_{net}$ values but not all GCMs converge towards significant changes in this region. Moreover, the model evaluation showed a lower performance over this area (Fig. 1a). For these two reasons, the results may be less certain.

For the year 2050, Schadlach et al. (2013) estimated an average increase of 70 mm year$^{-1}$ (+15% compared to the baseline of

500 470 mm year$^{-1}$) of the $I_{net}$ over the European continent under a high emission scenario. For the same scenario, Fischer et al. (2007) predicted an increase of 53 mm year$^{-1}$ (+36% compared to the baseline of 147 mm year$^{-1}$) over western Europe by 2080. These values are comparable to the results of this study, presenting a mean increase of 38 mm year$^{-1}$ (+17% compared to the baseline of 221 mm year$^{-1}$, integrated over the three summer months only) and 67 mm year$^{-1}$ (+30%) for 2050 and 2090, respectively, and for a comparable emission scenario (SSP3-7.0). Also, the evolution of $I_{net}$ under SSP5-8.5 (+35%) can be

related to the findings from Wada et al. (2013) for which increases larger than 25% are estimated over almost the entire European continent. Under this same scenario, the results show that France will face extreme increases in $I_{net}$ (+75%) confirmed by Fader et al. (2016) (+80%). Smaller increases were found in other studies (e.g., Döll, 2002; Elliott et al., 2014), however absolute values are hard to directly confront to literature because of the differences in methodology. In literature, $I_{net}$ is often

assessed under the assumption of potential irrigation during the entire year or growing season (as opposed to the summer only in this study), considering other factors such as irrigation efficiencies and strategies, varying crop types, and even population increase or economic growth ultimately impacting e.g. irrigation efficiencies. Furthermore, Elliott et al. (2014) demonstrated that irrigation estimates from GHMs deviate strongly from crop model predictions, potentially due to the differences in agrohydrological processes between the two types of models. Also, Wada et al. (2013) proved that the largest part of uncertainty in future $I_{net}$ estimation is due to the impact model in the first place, and only then to climate uncertainty. $ET_0$ is a determinant factor for these kinds of studies, and its calculation procedures can have an important influence on the final results (Webber et al., 2016).

Atmospheric data alone could give an indication of the crop water requirement, as is done in meteorological drought assessments. However, the integration of P and $ET_0$ into a crop model with application of irrigation is more realistic to estimate $I_{net,}$ because it benefits from the land system memory. It should be noted though that the wetness of the irrigated land area will in turn affect turbulent fluxes and thus atmospheric variables in general (Hirsch et al., 2017; Thiery et al., 2017; 2020; Keune et al., 2018). This feedback loop is not included in the presented simulations and needs to be carefully considered in future attempts to design climate-smart irrigation systems.

Whereas the focus of this study was on the irrigation requirement, a similar analysis can be performed in terms of agricultural productivity. An increase in $I_{net}$ is expected but, following increasing $CO_2$ concentrations, biomass production is also expected to increase (Schleussner et al., 2018; Vanuytrecht, 2020; Vanuytrecht et al., 2012). The yield water productivity ($WP_{Y/ET}$, i.e. the ratio between crop yield and the amount of water lost by evapotranspiration) will improve due to the rising $CO_2$ concentrations. Since crops can only fully profit of the $CO_2$ fertilization when soil fertility is high (Raes et al., 2021), an increase in $WP_{Y/ET}$ is likely to occur in irrigated fields that are generally well fertilized. In the absence of soil water and soil fertility stress, crop production might increase by about 25% up to 45% for an atmospheric $CO_2$ concentration of 550 ppm (Raes et al., 2017). Effects above this concentration remain more uncertain.

## 5.3 Future adaptations of irrigation infrastructure and management

Different practical future pathways can be considered starting from the current state of irrigation requirement. In regions where $I_{net}$ is currently low (low baseline $I_{net}$), there is typically no irrigation infrastructure available or needed to achieve a fairly high crop production. However, to maintain crop production in the future, large investments will be required to develop or extend the irrigation infrastructure (Rosa et al., 2020). In regions with an existing water shortage and irrigation infrastructure, the focus will be on improving irrigation efficiencies, aiming to buffer the effects of climate change (Jägermeyr et al., 2016). Our study did not consider specific irrigation practices and efficiencies. The latter have been estimated around 50% in Europe (Fischer et al., 2007; Rohwer et al., 2007; Wada et al., 2013), meaning that the $I_{net}$ values presented in this study could roughly be doubled to obtain the gross requirement. Sprinkler irrigation remains the most widely used practice in Europe, but the share of drip irrigation is progressively increasing in southern countries such as Spain and Italy (Monaghan et al., 2013), aiming at improving irrigation efficiencies. Furthermore, with a lower availability of freshwater, the introduction of other irrigation

strategies, such as deficit irrigation, also gains importance. Deficit irrigation intends to maximize crop water productivity, therefore stabilizing crop yields through time (Geerts and Raes, 2009; Mushtaq and Moghaddasi, 2011).

## 5.4 Model uncertainty

Model uncertainty is an important factor influencing climate scenario analyses (Lehner et al., 2020). This starts with the high variability between climate scenarios that are input to the crop model simulations. The uncertainty of future climate was included by using meteorological input from three scenarios and five GCMs, resulting in 15 different SSP-GCM combinations. The process of using only a small fraction of the various existing GCMs has been criticized (McSweeney and Jones, 2016). However, previous drought and irrigation projections often used less than five GCMs or used more but for only one emission

scenario. Additionally, the ISIMIP GCMs are carefully selected to represent the entire CMIP ensemble (Frieler et al., 2017; Warszawski et al., 2014).

The AquaCrop model setup also adds uncertainty. First, the constantly evolving field practices in terms of e.g. crop type and cultivars, water management, and soil fertility management were not included in the model simulations. However, this aspect is almost impossible to include. Second, the model generalizations (generic C3-type of crop, unconstrained water availability

and constant small soil fertility stress for the whole domain) increase the uncertainty in the projections. It should be noted that actual area of irrigated land is not considered, and consequently, the expansion thereof is not simulated (estimated by e.g. Schaldach et al., 2012). Nevertheless, the intention of this study is to limit the uncertainty in time and space (as described in section 2.2) by assuming these generalizations, therefore aiming to present in a simple way the evolution of $I_{net}$ during summer months.

## 6 Conclusions

Large-scale AquaCrop simulations over Europe were performed using ISIMIP3 meteorological forcings at a spatial resolution of 0.5° lat x 0.5° lon to assess future changes in net irrigation requirements $I_{net}$. Because this is the first large-scale AquaCrop application with ISIMIP3 input, the model was first evaluated using satellite-based SSM. The reanalysis-driven (ISIMIP3a) simulated SSM have a mean spatial ubRMSD of 0.06 $m^3m^{-3}$ with SMAP retrievals, and thereby deviate slightly more than the

assumed intrinsic error of the satellite retrieval error (0.04 $m^3m^{-3}$). The performance of AquaCrop compared with SMOS (ubRMSD=0.08 $m^3m^{-3}$) is slightly lower than with SMAP, most likely because the SMOS sensor suffers more from radio frequency interference. When using GCM-driven (ISIMIP3b) meteorology as input, the multi-year average SSM of the simulations is comparable to that of reference satellite data (ubRMSD=0.03 $m^3m^{-3}$), which reinforces the reliability of the ISIMIP3 climate data for future projections.

In the second part of this paper, the summer irrigation requirement of a near- (2031-2060) and far- (2071-2100) future horizon was simulated using five different GCMs and three emissions scenarios. We present net irrigation requirement values that are independent of the irrigated area, period, infrastructure and the exact crop type. The mean and interannual variability in net

irrigation requirement $I_{net}$ for the summer months were quantified for the two future climate horizons and compared to the baseline period (1985-2014). This evaluation showed that the effect of climate change on future $I_{net}$ depends on the emission scenario, but more strongly on the GCM. Under high and extreme emission scenarios (SSP3-7.0 and SSP5-8.5), almost the whole European continent will see an increase in summer $I_{net}$, with on average 30% and 35% additional net irrigation water required in the far future relative to the baseline $I_{net}$. Especially regions with a moderate baseline $I_{net}$ will experience strong increases in $I_{net}$. All GCMs agree on significant increases in central to southern Europe, which is in line with meteorological and soil moisture drought projections for the same scenarios, as well as previous irrigation demand projections.

The interannual variability in summer $I_{net}$ was quantified by the range between maximum and minimum $I_{net}$ within the 30-year climate periods, $RI_{net}$. It was found that mild increases in $I_{net}$ result in larger gaps between maximum and minimum summer $I_{net}$ within a time window, corresponding to more extremes, and a high interannual variability (large $RI_{net}$). In the future, northern and central areas will face increased $RI_{net}$, whereas southern Europe is likely to see the variability diminish resulting in steady high $I_{net}$. Under the strong mitigation scenario (SSP1-2.6), $I_{net}$ stabilizes towards the end of the century, consistent with the plateauing $CO_2$ concentrations in this scenario. The increase in variability is also reduced under this scenario. Overall, extra water will be required, but more production can be achieved under higher $CO_2$ concentrations. The exact effect of $CO_2$ fertilization remains uncertain, but it is expected that yield, and especially yield water productivity, are likely to increase in the future in absence of water and soil fertility stress. Our large-scale setup with AquaCrop is well suited to explore the effect of climate scenarios on crop productivity in future research.

These results highlight the importance of climate change mitigation to keep future irrigation at reasonable levels, while it also stresses the high uncertainty of climate projections. This study aimed to demonstrate the effect of climate change on $I_{net}$ over Europe, without considering land use, crop types, and actual irrigated areas to avoid the inclusion of more uncertainty. Therefore, the results of this study should not be taken as predictions but as an indication of the potential consequences of climate change on the amount and variability of $I_{net}$ for the summer months.

**Appendices**

**Appendix A: comparison of SIM1 and SMOS SSM in terms of anomalies**

To evaluate the short-term and interannual variability in the AquaCrop SSM in terms of temporal anomaly correlation (anomR), time series of anomalies are calculated by subtracting the climatology from simulation and satellite data for each daily time step. The climatology calculates the mean seasonal cycle as a long-term mean using a sliding window of 31 days with a minimum threshold of three data points of data within the window.

Because anomR values can only be computed when several years of data are available, the AquaCrop SSM simulations forced with ISIMIP3a reanalysis data for the years 2011-2016 (SIM1) are evaluated against SMOS SSM retrievals only. Figure A1 shows the anomR over Europe with a spatial mean anomR of 0.44. Higher correlations are found in south-western locations

(anomR often > 0.6) and lower performances occur in north and central-western Europe (anomR generally < 0.4). Also shown on this figure is a partitioning of Europe in various zones for further discussion.

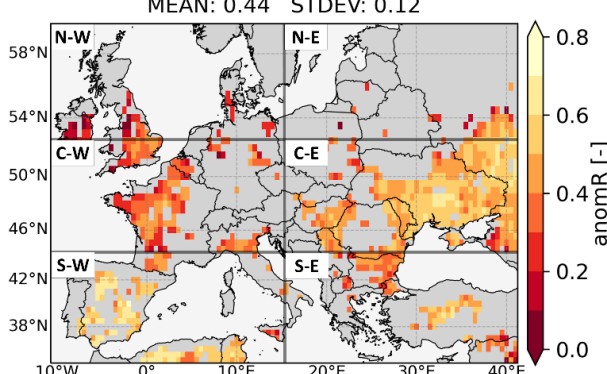

**Figure A1** Anomaly correlation (anomR) between SIM1 AquaCrop SSM and SMOS SSM for the period 2011-2016. The spatial mean and standard deviation are indicated (MEAN, STDEV). The six European sub-regions used to describe the model evaluation and the future $I_{net}$ are indicated (from top left to bottom right: north-west, north-east, central-west, central-east, south-west, south-east).

## Code and data availability

The regional AquaCrop (v6.1) is available on Zenodo at https://doi.org/10.5194/gmd-2021-98 (de Roos et al., 2021). The model setup of this specific research and the netCDF files of the net irrigation requirement projections are available on Zenodo at https://doi.org/10.5281/zenodo.6760977 (Busschaert et al., 2022). ISIMIP input data can be retrieved from https://www.isimip.org/gettingstarted/input-data-bias-correction/. SMAP L2 radiometer half-orbit 36 km EASE-Grid soil moisture version 7 is available at https://nsidc.org/data/SPL2SMP/versions/7 (O'Neill et al., 2020; https://doi.org/10.5067/F1TZ0CBN1F5N). SMOS L2 soil moisture version 650 can be downloaded at https://smos-diss.eo.esa.int/socat/SMOS_Open/.

## Author contributions

LB adapted the code to run the regional version of AquaCrop with ISIMIP data, prepared the input data, conducted the model evaluation, performed all simulations and analyses. SdR provided the code of the regional version of AquaCrop (v6.1) and scientific guidance. GDL prioritized the main steps taken in the paper, provided supervision and scientific guidance throughout all research advances, and manages HPC usage. WT provided scientific guidance through climate change impact assessments and ISIMIP. DR provided scientific guidance regarding the use and interpretation of AquaCrop, along with an appropriate methodology to assess the future irrigation requirement. LB wrote the paper and all authors contributed.

## Competing interests

The authors declare that they have no conflict of interests.

## Acknowledgements

The resources and services used in this work were provided by the VSC (Flemish Supercomputer Center), funded by the Research Foundation - Flanders (FWO) and the Flemish Government. The authors would like to acknowledge the ISIMIP for providing climate input data used in this study. We would also like to thank Luke Grant for his help with the ISIMIP data downloads and storage management. The authors appreciate the constructive reviews from the two anonymous reviewers.

## Financial support

This research is conducted as part of the H2020 project Shui, that stands for "*Soil Hydrology research platform underpinning innovation to manage water scarcity in European and Chinese cropping systems*". SHui is funded by the European Union Project GA 773903. Additional support was available via KU Leuven internal fund C14/21/057.

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
