# Peer review of "Net irrigation requirement under different climate scenarios using AquaCrop over Europe"

_Hydrology and Earth System Sciences, 2021_

## Author Comment (AC1)

The authors thank the reviewer for the valuable comments and recommendations to improve our manuscript. The comments of the reviewer are in boldface, whereas our responses are in normal fonts. Italic text is quoted from the updated manuscript. Line numbers refer to the manuscript version with track changes.

**Reviewer #1:**

1. **The article 'Net irrigation requirement under different climate scenarios using Aquacrop over Europe' quantifies the current net irrigation and the future variations under different climate change scenarios for a single crop all over Europe. The topic has great potential, and it is particularly up to date and interesting to be analyzed, however, some hypotheses and limitations strongly affect the potential of the results here presented. The major concerns are related to the choice of a unique crop type to be simulated all around Europe. Even if a strong statistical analysis is beyond the reported calculation, it is completely missing how the choice of this crop type is significative for Europe. Are C3 the most widespread crops in Europe?**

Answer: The paper now includes a paragraph to explain that the irrigation requirement is computed independent of the irrigation area, period and exact crop type. This paragraph was added to section 2.2, it now better explains these generalizations.

L134-136:

*"Climate impact assessments are subject to large uncertainties, which increase with longer temporal projections. Therefore, several assumptions are made in this study to limit the uncertainty of other factors than climate. We will present net irrigation requirement values that are independent of the irrigated area, period, infrastructure and the exact crop type."*

L147-149:

*"For future projections, the use of a representative field crop is supported by the current lack of detailed year- and location-specific crop maps, and by the unpredictability of changes and developments in crop type and distribution."*

Regarding the choice the crop type, we have now clarified that two different crops are used in this study: (i) the general C3 crop (from de Roos et al., 2021) for the historical evaluation of the model; and (ii) a representative field crop to assess future $I_{net}$.

For the historical evaluation, we added a statement about the fact that C3 crops are dominant in Europe.

L173-174:

*"C3 crops are dominant in Europe (Monfreda et al., 2008; Still et al., 2003)."*

For the $I_{net}$ projections, the 'modified generic C3 crop' is indeed a poor and misleading identification of the crop with which the net irrigation requirement ($I_{net}$) was determined. Therefore, this "unique" crop type is now referred to in the paper as a "representative field crop". It has the typical characteristics of most field crops cultivated in Europe. A table and some information about the considered crop characteristics have been added to the paper. If the crop is a C3 or C4 is indeed important when determining the crop yield. However, for determining $I_{net}$, only the crop characteristics that determine the evapotranspiration need to be considered, i.e. the crop transpiration coefficient, the canopy cover, and the thresholds for water and temperature stress.

**Table 1.** Characteristics of the representative field crop

| Crop parameters | values | units |
|---|---|---|
| Canopy cover (during the 3 summer months) | 85 | % soil cover |
| Crop transpiration coefficient when canopy is complete | 1.10 | - |
| Maximum effective rooting depth | 1.0 | m |
| Soil water depletion at which stomata starts to close | 50 | %TAW |
| Base temperature, below which crop development does not progress | 8.0 | °C |
| Minimum growing degrees required for full crop transpiration | 10.0 | °C-day |

The following text was added to describe this representative field crop:

L179-186:

*"For the determination of $I_{net}$, a representative field crop is considered. The crop characteristics that determine crop transpiration and hence $I_{net}$ are listed in Table 1. The considered crop transpiration coefficient of 1.10 is a good indicative value of the basal crop coefficient for the mid-season for a large range of field crops (Allen et al., 1998). Moreover, it is assumed that in the summer months (in which $I_{net}$ is determined) the crop has reached its maximum canopy cover and is prior to senescence. Since $I_{net}$ is determined by keeping the soil water content in the root zone above 50 % of the readily available water (RAW, which is 25 % of the total available water, TAW, for the representative field crop), water stress does not affect crop transpiration. Also, air temperature stress affecting crop transpiration will be small or absent in the summer months with the settings of the thresholds in Table 1."*

2. **Similarly, in the method it is not clearly stated on which areas the calculation has been computed.**

Answer: The purpose of this study is to quantify irrigation amounts required for optimal crop growth, without including explicit future projections on the extent of irrigated areas (such as available through e.g., Hurtt et al., 2020). This has been clarified in the newly added paragraph of section 2.2.

L136-140:

*"First, simulations are performed over all pixels of the entire study domain (i.e. the main European continent), and the irrigation estimates for the entire hypothetically irrigated agricultural domain are normalized by area to make the results independent of the actual irrigated area. This avoids the need to include estimates of future hypothetical land use (Prestele at al., 2016), and the uncertain evolution of the extent of irrigated areas (Schaldach et al., 2012; Hurtt et al., 2020)."*

3. **Secondly, the spatial resolution of the analysis is coarse, since some global assessment of water for irrigation works at 10km at the equator and maps of harvested areas are available at a resolution up to 250m or even 30m.**

Answer: Our values of our irrigation estimates are generated at a 0.5° x 0.5° spatial resolution due to the spatial resolution of the meteorological forcing, and particularly the future climate scenarios. We further added the following text to highlight the distinction between our climate-based study and fine-scale agricultural studies.

L140-146:

*"Second, the spatial resolution of this study matches that of the ISIMIP input data resolution. In contrast to fine-scale agricultural studies, usually assessing actual irrigation under historical conditions, future climate projections are dependent on the resolution of the driving climate models*

*(or downscaled output). Such studies mainly aim at estimating the irrigation requirement that is needed for crop root uptake, thereby omitting the part of irrigation that is lost to the atmosphere, or retained on the soil surface or in the soil profile. Also, state-of-the-art global and continental-scale climate impact assessments are typically performed at the same resolution (e.g., Jägermeyr et al., 2021; Lange et al., 2020; Thiery et al., 2021)."*

4. **I furtherly suggest that these works are seen and cited in the article (Liu et al., 2010; Siebert and Doll, 2010, Chiarelli et al., 2020).**

Answer: These are indeed relevant studies, thank you for mentioning them. They were carefully considered and included in the paper.

5. **Third, some strong assumptions are related to crop characteristic and growing period. While only one crop has been selected, results are reported for only the three summer months, when it could be expected that in future more water is needed even outside the summer months. Furtherly, farmers can adopt different techniques, shifting to different crops or even adapting the planting and harvesting period accordingly to the new climatic condition. This latter option is not mentioned in the paper, neither discussed, while it could greatly influence the reported results.**

Answer: We agree with the reviewer on this point. As mentioned in the comment, shifts in the growing period can be expected to follow the new climatic conditions but, ultimately, this remains a decision taken by the farmer in the context of climate change adaptation, and characterized by uncertainty. Therefore, we decided to present results integrated over the summer months only (current growing periods in Europe) and to assume no climate change adaptation (crop type, growing season, fertilizer use, etc.), to make the irrigation amounts directly comparable to the baseline. We added the following text in this extra paragraph (section 2.2) to support this choice.

L149-155:

*"Finally, the uncertainty and high spatial and temporal variability of the start and end of the growing season (King et al., 2018; Menzel and Fabian, 1999; Schadlach et al., 2012) restricts the modelling possibilities. Some previous studies (e.g., Elliott et al., 2014; Fader et al., 2016; Fischer et al., 2007; Konzmann et al., 2012) have used dynamic growing seasons, but the choice has been made to avoid this additional level of uncertainty for this study. Therefore, only the summer months are considered to make the future requirement directly comparable to the baseline $I_{net}$. On average, these are the months presenting the highest $I_{net}$ (Siebert and Döll, 2010), and are expected to remain important months for irrigation requirements, even if growing seasons might shift in the future."*

For clarity, we now also express the results in mm month$^{-1}$ (instead of mm year$^{-1}$), and specify that the values are computed for the summer months only.

6. **The quite lengthy description of the statistical analyses obfuscates the main outcomes of the paper regarding the variation in the irrigation demand, potentially distracting the reader from the most interesting results. I suggest moving some of the analysis as supplementary material for better highlighting the effects of climate change scenarios in the different regions of Europe.**

Answer: We moved the SIM1 evaluation in terms of anomalies (originally in section 4.1.1) to the Appendix. Also, the comparison of SIM2 with SIM1 (section 4.1.2) was removed to have consistent comparison periods (i.e. the period of jointly available SMAP and SMOS data, from April 2015 onwards). We realize that the model evaluation constitutes an important part of the paper, but we would like to highlight the importance of the two types of evaluation. The analysis performed with

SIM1 is a short-term evaluation with reanalysis meteorology whereas SIM2 constitutes a climatological (long-term) evaluation with GCM meteorology. Both are important analyses to understand the model performance and the GCMs for historical periods. This is clarified in the text and in the new Table 2, describing the different simulations (replacing former Fig. 1).

L290-293:

*"Both the time series of historical SIM1 and SIM2 SSM are compared to satellite observations through the skill metrics described in section 3.1.1 for the time period with available data for both SMOS and SMAP, i.e. from April 2015 through 2016. SIM1 is a short-term evaluation since daily SSM simulations are compared to satellite observations."*

L295-298:

*"For the historical SIM2 SSM simulations, the multi-year average (long-term) results driven by the five different GCMs, and the median SSM time series across the GCMs are evaluated."*
* * *
Minor comments:

**7. Line 110: Please, better state which can be the advantages of the results your presented in this paper.**

Answer: The advantages are now clarified in the introduction.

L117-124:

*"Compared to previous studies, the advantages are that the simulations are performed with (i) climate data from the latest generation of reanalyses and GCMs, (ii) the most recent set of future scenarios, and (iii) a crop model (AquaCrop), in which the dynamic interactions between water and vegetation are the main focus and where irrigation and management practices can be included with more detail than in a land surface or hydrological model. Future $I_{net}$ projections could be used to inform on climate change adaptation strategies (e.g., climate-smart irrigation, crop type selection, water conservation). The new AquaCrop-ISIMIP3 model setup can be run at any spatial domain and resolution, providing future opportunities for further climate analysis, also including other irrigation practices and management options."*

**8. Line 127: 1m of soil depth is your assumption?**

Answer: We understand that the description of the soil profile may have been confusing. Therefore, we added an extra sentence to clarify this.

L165-167:

*"A total profile depth of 1.30 m is defined, but without the presence of a groundwater table or confining layers, the actual depth below the maximum rooting depth has no influence on the simulations."*

**9. Method: on which areas has the model run?**

Answer: This has now been clarified by describing the hypotheses in the new paragraph of section 2.2 (see R1#2).

**10. Line 194: Your hypothesis?**

Answer: For optimal irrigation scheduling, irrigation water is applied when the root zone depletion is 100% RAW (timing of irrigation). The amount of water applied (amount of irrigation) will bring back the root zone to field capacity (which corresponds with 0% RAW depletion). Hence during the irrigation interval, the root zone depletion varies between 0 and 100 % RAW depletion. The 50% root zone depletion is hence the average root zone depletion in the irrigation interval.

The text has been adjusted in the manuscript.

L254-256:

*"By selecting a threshold of 50% RAW depletion, which is the average depletion in an optimal irrigation interval (Smith, 1992), crop water stress affecting the canopy development and transpiration of the representative field crop is avoided, and effective rainfall (the part stored in the root system up to field capacity) is still considered."*

**11. Line 208: Is the volumetric content an average on the entire pixel? Is this comparable with the result of a single crop?**

Answer: Indeed, soil moisture values are averages over an entire pixel. All input and AquaCrop variables are integrated over one pixel. The following sentence has been added for clarity.

L146-147:

*"Third, each pixel is defined as a hypothetical homogeneous field, in which the vegetation conditions are identical."*

**12. Line 234: How are initial simulation conditions set?**

Answer: Thank you for noticing this information was missing. Information about the spinup has now been added to the manuscript.

L241-243:

*"SIM1 and SIM2 have a spin up period of four years, and only output from 2015 onwards is used for evaluation, i.e. starting when both SMOS and SMAP data are available."*

L248-249:

*"For the baseline simulation, the initial moisture conditions are set to field capacity while the future periods have a spin up of at least 10 years (continuous simulation from 2021 through 2100)."*

**13. Results: Are you referring to consumptive water use or water withdrawals?**

Answer: The net irrigation requirement concept is explained in the following text.

L249-253:

*"For SIM3, irrigation is introduced, using the net irrigation requirement option in AquaCrop, whereby a small amount of water (just covering the crop ET for that day) is injected into the root system on days when a certain fraction of the RAW is depleted (Raes et al., 2017). With this option, solely the amount water taken up by the roots is considered, where the wetting of the soil surface, and interval and application amount specific to a particular irrigation method are not relevant."*

$I_{net}$ could therefore not be categorized as consumptive water (since it is not equal to the total evapotranspiration), nor as water withdrawals (since the irrigation water added is not withdrawn from a system).

---

## Author Comment (AC2)

The authors thank the reviewer for the valuable comments and recommendations to improve our manuscript. The comments of the reviewer are in boldface, whereas our responses are in normal fonts. Italic text is quoted from the updated manuscript. Line numbers refer to the manuscript version with track changes.

**Reviewer #2:**

1. *Novelty*: **Several studies on irrigation water requirements today and in the future have been published since the first study by Döll et al 20 years ago. The study by Busschaert et al. (2022) would have been a welcomed addition to the literature 20 years ago, but in 2022 I am struggling to see where Busschaert et al. (2022) provides new insights in the scientific area. Several single model studies and multiple model studies have been presented, not the least coming out of ISIMIP, so what does this paper include that isn't well covered in the existing literature? My recommendation is that a more convincing presentations of why this study is needed must be included, if this paper is to be published in HESS.**

Answer: Thank you for highlighting this point. The novelty of this study has now been clarified in the following text.

L117-124:

*"Compared to previous studies, the advantages are that the simulations are performed with (i) climate data from the latest generation of reanalysis and GCMs, (ii) the most recent set of future scenarios, and (iii) a crop model (AquaCrop), in which the dynamic interactions between water and vegetation are the main focus and where irrigation and management practices can be included with more detail than in a land surface or hydrological model. Future $I_{net}$ projections could be used to inform on climate change adaptation strategies (e.g., climate-smart irrigation, crop type selection, water conservation). The new AquaCrop-ISIMIP3 model setup can be run at any spatial domain and resolution, providing future opportunities for further climate analysis, also including other irrigation practices and management options."*

2. *Model setup*: **The authors have chosen to perform the study at a fairly coarse spatial resolution, and with only one crop type all over Europe. I agree with what reviewer #1 has said on the choice of crop type, and the fact that you analyze summer months only. These seems like questionable and outdated decisions.**

Answer: We understand that some elements in the applied methods were insufficiently clarified and motivated in the original submission. Therefore, we added an extra paragraph (in section 2.2) to explain and support our assumptions.

L134-155:

*"Climate impact assessments are subject to large uncertainties, which increase with longer temporal projections. Therefore, several assumptions are made in this study to limit the uncertainty of other factors than climate. We will present net irrigation requirement values that are independent of the irrigated area, period, infrastructure and the exact crop type. First, simulations are performed over all pixels of the entire study domain (i.e. the main European continent), and the irrigation estimates for the entire hypothetically irrigated agricultural domain are normalized by area to make the results independent of the actual irrigated area. This avoids the need to include estimates of future hypothetical land use (Prestele at al., 2016), and the uncertain evolution of the extent of irrigated areas (Schaldach et al., 2012; Hurtt et al., 2020). Second, the spatial resolution of this study matches that of the ISIMIP input data resolution. In contrast to fine-scale agricultural studies, usually assessing actual*

*irrigation under historical conditions, future climate projections are dependent on the resolution of the driving climate models (or downscaled output). Such studies mainly aim at estimating the irrigation requirement that is needed for crop root uptake, thereby omitting the part of irrigation that is lost to the atmosphere, or retained on the soil surface or in the soil profile. Also, state-of-the-art global and continental-scale climate impact assessments are typically performed at the same resolution (e.g., Jägermeyr et al., 2021; Lange et al., 2020; Thiery et al., 2021). Third, each pixel is defined as a hypothetical homogeneous field, in which the vegetation conditions are identical. For future projections, the use of a representative field crop is supported by the current lack of detailed year- and location-specific crop maps, and by the unpredictability of changes and developments in crop type and distribution. Finally, the uncertainty and high spatial and temporal variability of the start and end of the growing season (King et al., 2018; Menzel and Fabian, 1999; Schadlach et al., 2012) restricts the modelling possibilities. Some previous studies (e.g., Elliott et al., 2014; Fader et al., 2016; Fischer et al., 2007; Konzmann et al., 2012) have used dynamic growing seasons, but the choice has been made to avoid this additional level of uncertainty for this study. Therefore, only the summer months are considered to make the future requirement directly comparable to the baseline $I_{net}$. On average, these are the months presenting the highest $I_{net}$ (Siebert and Döll, 2010), and are expected to maintain important months for irrigation requirements, even if growing seasons might shift in the future."*

For clarity, we now also express the results in mm month$^{-1}$ (instead of mm year$^{-1}$), and mark that the values are computed on the summer months only.

***Evaluation – comparison to satellite observations*:**

3. **The evaluation (comparison to satellite observations) are performed at different timescales for SIM1 and SIM2, as you mention. However, if you are going to compare the comparisons, I think you must try to make them a little more comparable, e.g. by including similar comparisons for SIM1 as you have for SIM2. Also, and what really puzzles me, is that as far as I understand, for SIM1, no irrigation is included in the simulations, whereas in SIM2 it is. These simulation results are both compared to satellite estimates, and both are deemed "reasonable". Consequently, I will argue that irrigation doesn't really impact your evaluations much, is that true? What does this say about your irrigation water use and your evaluations? How does the coarse scale and your irrigated areas impact the evaluation results? In any case, I think you must do both comparisons with the same model setup.**

Answer: We agree that the description of the three types of simulations may have been unclear in the original submission. SIM1 and SIM2 are evaluation simulations and are both run without irrigation. SIM3 constitute a set of simulations intended to make $I_{net}$ projections, and are run with irrigation turned on. The characteristics and purpose of each simulation type have now been clarified in the new Table 2 (below, replacing former Fig. 1). We hope this makes the simulation setup and assumptions clearer.

**Table 2** Description of the different simulations with regard to the simulation periods, analyses, climate data, crop characteristics, soil fertility stress and the activation of the irrigation (ON/OFF).

| Sim. | Period | Analysis | Climate data | Crop | Soil fertility stress (%) | Irrigation |
|------|--------|----------|--------------|------|---------------------------|------------|
| **SIM1** | April 2015 - December 2016 | SSM short-term evaluation | reanalysis | generic C3 | 30 | OFF |
| **SIM2** | April 2015 - December 2020 | SSM climatological evaluation | 5 GCMs | generic C3 | 30 | OFF |
| **SIM3** | 1985-2014 2031-2060 2071-2100 | $I_{net}$ projections | 5 GCMs 5 GCMs x 3 SSPs | representative field crop | 20 | ON |

Furthermore, the evaluation time periods have been adapted for consistency, as suggested by the reviewer. SIM2 SSM climatologies are now compared to satellite SSM climatologies for consistent time periods (i.e. from April 2015 through 2020).

About the influence of actual irrigation on the model, an analysis was performed (results are not shown in the manuscript) and discussed in the last paragraph of section 5.1. As explained, we did not find lower correlations between AquaCrop simulated SSM (run without irrigation) for pixels presenting a high percentage of area equipped for irrigation. The possible reasons for these results were explained in the original manuscript, and an reference has been added to support our findings.

L537-538:

*"[…] and a recent study stressed the low potential to detect irrigation at coarse resolutions (Dari et al., 2021)."*

4. **Also, for the evaluation of results using GCM input you state that you use climatology for 6-10 years, depending on the data product. I would argue that for GCMs, nether 6 nor 10 years is not enough to call it "climatology", and your results can be very much impacted of what happens in that 10-year period, and I do not think you should rule "reasonable" on the background of 6-10 years. Any thoughts?**

Answer: We understand this comment, and we agree that robust climatologies should be computed on longer time periods. However, we are limited by the availability of satellite data (SMAP SSM data is available from April 2015 onwards) for the evaluation period. For the same reason, other studies and applications (e.g., satellite data assimilation systems) are also computing climatologies on relatively short time periods. We added the following statement to support this.

L303-306:

*"The computation of the climatology is restrained to the availability of reference satellite data (i.e. SMAP, data available from April 2015), as it is also the case in satellite data assimilation systems (e.g., SMAP Level 4 product; Reichle et al., 2019)."*

5. **SIM1 and SIM2 are evaluated in very different ways, and I find it problematic that you still put the evaluations in the same figure and to some extent compare them in the text. Are you comparing apples and potatoes, and how can you make the evaluations and analyses more consistent?**

Answer: We agree that SIM1 and SIM2 should not be directly compared to each other. We therefore removed this comparison in the text and in the new Fig. 2 and 3 (replacing Fig. 4 and 5 from the original manuscript).
* * *
Minor issues:

***Clarifications needed*:**

**6. I think you only model agricultural areas, and specifically agricultural areas that are irrigated, is that right? I can't, however, see that you refer to any dataset used to define these areas?**

Answer: Since the future extent of irrigated areas is uncertain, we decided to simulate cropland across the entire domain (i.e. on every land grid cell). Irrigation estimates are normalized by grid cell area to make results independent of the actual irrigated area. Please see the answer to the comment R2#2 presenting this extra paragraph (L134-155).

**7. Also, if I understand this right, you simulate the entire cells at 0.5x0.5 degrees with the C3 crop type. Do you account for this when comparing to satellite observations, or are the soil moisture estimates performed point by point without regarding how much is actually irrigated (see also above)? If you do not account for partially irrigated cells, don't you think that you overestimate soil moisture in many cells?**

Answer: As mentioned in R2#6, we do not account for partially irrigated cells. For the evaluation simulations (SIM1 and SIM2), irrigation is not activated and the SSM is evaluated against satellite retrievals for all available (after screening bad quality retrievals, see section 2.4) data retrievals. Soil moisture is therefore not overestimated by AquaCrop. But it could have been underestimated in areas where irrigation is actually applied. Please see the answer to the comment R2#3 about the discussion on the impact of actual irrigation on the model performance.

**8. *Evaluation*: Do you compare nearest neighbour of the satellites and simulation cells, or how is it done?**

Answer: Indeed, nearest-neighbour sampling is used to compare AquaCrop simulated SSM to satellite retrievals. This is mentioned in the following sentence from the original manuscript.

L268-269:

*"To compare the spatial and temporal patterns of SSM from 0.5° AquaCrop simulations with 36-km satellite data, nearest-neighbour sampling is used to spatially match simulated SSM with SMOS and SMAP retrievals."*

**9. *Irrigation method*: You inject water into the root system. Do you know how large part of the European irrigation is this efficient? Possibly add some thought on choice of irrigation method and how it may influence your results?**

Answer: This is a very interesting suggestion. The following text has been added to reflect on this.

L590-595:

*"Our study did not consider specific irrigation practices and efficiencies. The latter have been estimated around 50% in Europe (Fischer et al., 2007; Rohwer et al., 2007; Wada et al., 2013), meaning that the $I_{net}$ values presented in this study could roughly be doubled to obtain the gross requirement. Sprinkler*

*irrigation remains the most widely used practice in Europe, but the share of drip irrigation is progressively increasing in southern countries such as Spain and Italy (Monaghan et al., 2013), aiming at improving irrigation efficiencies."*

***Discussion***:

**10. I miss some thoughts on your irrigation water use estimates compared to already published estimates. You write that comparisons are difficult, which I agree on, but I still think it should be included.**

Answer: Thank you for this suggestion, we added a section including some comparisons (in absolute terms, and percentages).

L553-561:

*"For the year 2050, Schadlach et al. (2013) estimated an average increase of 70 mm $year^{-1}$ (+15% compared to the baseline of 470 mm $year^{-1}$) of the $I_{net}$ over the European continent under a high emission scenario. For the same scenario, Fischer et al. (2007) predicted an increase of 53 mm $year^{-1}$ (+36% compared to the baseline of 147 mm $year^{-1}$) over western Europe by 2080. These values are comparable to the results of this study, presenting a mean increase of 38 mm $year^{-1}$ (+17% compared to the baseline of 221 mm $year^{-1}$, integrated over the three summer months only) and 67 mm $year^{-1}$ (+30%) for 2050 and 2090, respectively, and for a comparable emission scenario (SSP3-7.0). Also, the evolution of $I_{net}$ under SSP5-8.5 (+35%) can be related to the findings from Wada et al. (2013) for which increases larger than 25% are estimated over almost the entire European continent. Under this same scenario, the results show that France will face extreme increases in $I_{net}$ (+75%) confirmed by Fader et al. (2016) (+80%)."*

**11. In the results part, you say you find decreases in Inet in locations along the Mediterrenean coast. To me this is surprising, and I think it deserves a sentence or two in the discussion. What are the underlying factors causing this?**

Answer: This statement concerns some regions that present historical extreme evapotranspiration rates, and insignificant rainfall in the summer months. Following the AquaCrop calculation procedures, the effect of stomatal closure by 5% as response to $CO_2$ concentrations above 550 ppm is stronger than the increase in ET for these regions (less than 5%). This has been clarified in the revised manuscript by adding the following text.

L410-413:

*"Decreases in $I_{net}$ (boxplot whiskers below 0, Fig. 4c) are only observed in a few southern coastal locations under the high and severe emission scenarios. In these historically warm and dry regions with insignificant rainfall in the summer months, the effect of stomatal closure by 5% as response to $CO_2$ concentrations above 550 ppm is stronger than the increase in ET (less than 5%)."*

**12. What can these results be used for? You touch the topic slightly in the discussion, but it would be favourable if you could link your results to adaptation issues somewhat closer.**

Answer: These results could be used to inform climate change adaptation strategies (e.g., climate-smart irrigation, crop type selection, water conservation). Also, the model setup itself can be run, with ISIMIP input data, to make projections at any resolution and over any study domain. This has now been clarified in the introduction. Moreover, further studies could elaborate on the impact of climate change on crop productivity, this has been added in the conclusion.

L121-124:

*"Future $I_{net}$ projections could be used to inform on climate change adaptation strategies (e.g., climate-smart irrigation, crop type selection, water conservation). The new AquaCrop-ISIMIP3 model setup can be run at any spatial domain and resolution, providing future opportunities for further climate analysis, also including other irrigation practices and management options."*

L645-646:

*"Our large-scale setup with AquaCrop is well suited to explore the effect of climate scenarios on crop productivity in future research."*

---

## Author Response (AR2)

The authors thank the reviewer for the valuable comments and recommendations to improve our manuscript. The comments of the reviewer are in boldface, whereas our responses are in normal fonts. Italic text is quoted from the updated manuscript. Line numbers refer to the manuscript version with track changes.

**Reviewer #2:**

1. Comparison to satellite observations: I am still a little puzzled as to what is compared here. The model simulations assume C3 crops all over, and no irrigation (SIM1 and SIM2). These simulation results are compared to satellite estimates, where soil moisture numbers are affected by whatever vegetation is present. Should the variability of these estimates necessarily follow each other? Would the results be similar with any vegetation type? I am just wondering if the comparison most of all say something about the quality of the precipitation variability in the ISIMIP datasets.

Answer: Meteorological forcings are indeed dominating the SSM variations but the aim is to evaluate the ability of AquaCrop to integrate these forcings and to provide SSM. With other types of vegetation, the simulated SSM would be biased but the main variations would remain unchanged (especially at the resolution of this study). For this reason, the model is also evaluated with bias-free metrics. This has been clarified at the end of section 3.1.1.

L265-269: "The aim of the historical evaluation is to assess the performance of AquaCrop to integrate ISIMIP3 meteorological forcings and to provide SSM estimates. By design, the model and satellite retrievals are biased, due to model parameters related to the soil and the uniform vegetation type (generic C3 crop), vertical representativeness bias, etc. Therefore, bias-free metrics (R and ubRMSD) are essential to assess whether the main temporal variations of SSM are captured by the model forced with ISIMIP3 data."

**Technical issues:**

2. *Europe:* Some northern Europeans (e.g. Finns) or Russians might feel a little left out by your definition of the European continent (line 124). Possibly you should state that you focus on the part of continental Europe that is between the chosen latitudes and longitudes.

Answer: Thank you for this recommendation. We adjusted this in the text.

L124-125: "The study domain focuses on the part of the European continent with latitudes (lat) ranging from 34.75° N to 59.75° N and longitudes (lon) from -10.75° E to 41.25° E."

**3. Possibly I missed something, but why are some areas in Fig. 1a grey? It would be helpful to explain in the figure caption or include in legend.**

Answer: Grey areas correspond to locations where there are not enough satellite retrievals to evaluate AquaCrop surface soil moisture. We clarified this in the caption of Fig. 1

L311-312: "Grey areas correspond to pixels where the number of satellite retrievals is smaller than 100."

The minimum threshold of observations per pixel was also specified in section 3.1.1.

L264: "A minimum threshold of N=100 reference data points in time are set per pixel for all analyses."